# Neural mechanisms of modulations of empathy and altruism by beliefs of others' pain

**Taoyu Wu, Shihui Han\***

School of Psychological and Cognitive Sciences, PKU-IDG/MGovern Institute for Brain Research, Beijing Key Laboratory of Behavior and Mental Health, Peking University, Beijing, China

**Abstract** Perceived cues signaling others' pain induce empathy which in turn motivates altruistic behavior toward those who appear suffering. This perception-emotion-behavior reactivity is the core of human altruism but does not always occur in real-life situations. Here, by integrating behavioral and multimodal neuroimaging measures, we investigate neural mechanisms underlying modulations of empathy and altruistic behavior by beliefs of others' pain (BOP). We show evidence that lack of BOP reduces subjective estimation of others' painful feelings and decreases monetary donations to those who show pain expressions. Moreover, lack of BOP attenuates neural responses to their pain expressions within 200 ms after face onset and modulates neural responses to others' pain in the insular, post-central, and frontal cortices. Our findings suggest that BOP provide a cognitive basis of human empathy and altruism and unravel the intermediate neural mechanisms.

## Introduction

Aesop's fable 'The boy who cried wolf' tells a story that villagers run or do not run to help a shepherd boy who cries wolf depending on whether or not they believe that the boy's crying indicates his actual emotion and need. This story illustrates an important character of human altruistic behavior, that is, perceived cues signaling others' suffering drives us to do them a favor only when we believe that their suffering is true. Although this character of human altruism was documented over 2000 years ago in Aesop's fable and is widely observed in current human societies, its psychological and neural underpinnings have not been fully understood. The present study investigated how beliefs of others' pain (BOP) modulate human altruistic behavior independently of perceived cues signaling others' suffering and whether the modulation effect, if any, is mediated by changes in empathy for others' pain and relevant brain underpinnings.

Empathy refers to understanding and sharing of others' emotional states (*Decety and Jackson, 2004*) and has been proposed to provide a key motivation for altruistic behavior in both humans and animals (*Batson et al., 2015*; *de Waal, 2008*; *Decety et al., 2016*). Empathy can be induced by perceived cues signaling others' pain that activate neural responses in brain regions underlying sensorimotor resonance (e.g., the sensorimotor cortex), affective sharing (e.g., the anterior insula [AI] and anterior cingulate cortex [ACC]), and mental state inference/perspective-taking (e.g., the medial prefrontal cortex [mPFC] and temporoparietal junction [TPJ]) (*Singer et al., 2004*; *Jackson et al., 2005*; *Avenanti et al., 2005*; *Saarela et al., 2007*; *Fan and Han, 2008*; *Shamay-Tsoory et al., 2009*; *Han et al., 2009*; *Sheng and Han, 2012*; *Fan et al., 2011*; *Lamm et al., 2011*; *Zhou and Han, 2021*). Neural responses to others' pain in the empathy network and functional connectivity between its key hubs can predict motives for subsequent altruistic actions (e.g., *Hein et al., 2010*; *Hein et al., 2016*; *Mathur et al., 2010*; *Luo et al., 2015*). These brain imaging findings revealed neural mechanisms underlying the perception-emotion-behavior reactivity (e.g., perceived pain-

**\*For correspondence:**
shan@pku.edu.cn

**Competing interests:** The authors declare that no competing interests exist.

empathy-help) that occurs often in everyday lives (*Eisenberg et al., 2010*; *Hofman, 2008*; *Penner et al., 2005*). However, empathic neural responses are influenced by multiple factors such as perceptual features depicting others' pain (*Gu and Han, 2007*; *Li and Han, 2019*), observers' perspectives and attention (*Gu and Han, 2007*; *Li and Han, 2010*; *Jauniaux et al., 2019*), and perceived social relationships between observers and empathy targets (*Xu et al., 2009*; *Avenanti et al., 2010*; *Hein et al., 2010*; *Mathur et al., 2010*; *Sheng and Han, 2012*; *Azevedo et al., 2013*; *Sheng et al., 2014*; *Sheng et al., 2016*; *Han, 2018*; *Zhou and Han, 2021*). What remains unclear is whether and how BOP modulates empathic brain activity through which to further influence altruistic behavior. To address these issues is crucial for understanding variations of empathy and altruism during complicated social interactions as illustrated in Aesop's fable.

Beliefs refer to mental representations of something that are not immediately present to the scenes but allow people to think beyond what is here and now (*Fuentes, 2019*). Beliefs reflect an organism's endorsement of a particular state of affairs as actual (*McKay and Dennett, 2009*). Beliefs that best approximate reality enable the believers to act effectively and maximize their survival (*Fodor, 1985*; *Millikan, 1995*). Previous research has shown that beliefs affect multiple mental processes such as visual awareness (*Sterzer et al., 2008*) and processing of emotions (*Petrovic et al., 2005*) including experiences of pain (*Wager et al., 2004*; *Colloca and Benedetti, 2005*). The function of beliefs is also manifested in increasing the efficiency of neural processes involved in decision-making and goal setting (*Garcés and Finkel, 2019*; *Régner et al., 2019*). Potential effects of beliefs on empathic neural responses were tested by presenting participants with photographs showing pain inflicted by needle injections into a hand that was believed to be or not to be anesthetized (*Lamm et al., 2007*). Functional magnetic resonance imaging (fMRI) of brain activity suggested modulations of insular responses to perceived pain by beliefs of anesthetization. However, the results cannot be interpreted exclusively by BOP because the stimuli (i.e., needles) used to induce beliefs of numbed and non-numbed hands were different. An ideal paradigm for testing modulations of empathy by BOP independently of perceived cues signaling others' pain should compare brain activities in response to identical stimuli under different beliefs and enable researchers to test how BOP influences altruistic behavior.

In six behavioral, electroencephalography (EEG), and fMRI experiments, the current study tested the hypothesis that BOP affects empathy and altruistic behavior by modulating brain activity in response to others' pain. Specifically, we predicted that lack of BOP may result in the inhibition of altruistic behavior by decreasing empathy and its underlying brain activity. Our behavioral, EEG, and fMRI experiments were designed based on the common beliefs that patients show pain expressions to manifest their actual feelings of pain whereas pain expressions performed by actors/actresses do not indicate their actual emotional states. To examine BOP effects on empathy, we experimentally manipulated BOP by asking participants to learn and remember different identities (i.e., patient or actor/actress) of a set of neutral faces during a learning procedure. Thereafter, we measured self-reports of others' pain and own unpleasantness from the participants when they viewed learned faces with pain or neutral expressions. During EEG/fMRI recording, the participants were asked to discriminate patient or actor/actress identities of faces with pain or neutral expressions. We compared self-reports of others' feelings and brain activities related to pain (vs. neutral) expressions of patients' faces with those related to actors/actresses' faces. If the perception of patients' pain expressions implicitly activates BOP whereas perception of actors/actresses' pain expressions does not activate BOP, we expected that lack of BOP (i.e., to compare actors/actresses vs. patients) would reduce self-report of empathy, empathic brain activity, and altruistic behavior. We further predicted that BOP effects on altruistic behavior might be mediated by decreased empathy and empathic brain activity due to lack of BOP.

Similar to previous research (*Jackson et al., 2005*; *Fan and Han, 2008*; *Hein et al., 2010*; *Mathur et al., 2010*; *Sheng and Han, 2012*), we adopted both subjective and objective estimations of empathy for others' pain. Subjective estimation of empathy for pain depends on the collection of self-reports of others' painful feelings and one's own unpleasantness when viewing others' suffering (e.g., *Bieri et al., 1990*; *Jackson et al., 2005*; *Lamm et al., 2007*; *Fan and Han, 2008*; *Sheng and Han, 2012*). Objective estimation of empathy for pain relies on recording of brain activities, using fMRI or EEG, that differentially respond to painful versus non-painful stimuli applied to others (e.g., *Singer et al., 2004*; *Jackson et al., 2005*; *Gu and Han, 2007*; *Fan and Han, 2008*; *Hein et al., 2010*) or to others' faces with pain versus neutral expressions (*Botvinick et al., 2005*; *Saarela et al.,*

*2007*; *Han et al., 2009*; *Sheng and Han, 2012*). Brain responses to perceived non-painful stimuli applied to others or neutral expressions were also collected to control empathy-unrelated perceptual or motor processes. fMRI studies revealed greater activations in the ACC, AI, and sensorimotor cortices in response to painful compared to non-painful stimuli applied to others (e.g., *Singer et al., 2004*; *Jackson et al., 2005*; *Gu and Han, 2007*; *Hein et al., 2010*; see *Lamm et al., 2011*; *Fan et al., 2011* for review). EEG studies showed that event-related potentials (ERPs) in response to perceived painful stimulations applied to others' body parts elicited neural responses that differentiated between painful and neutral stimuli over the frontal region as early as 140 ms after stimulus onset (*Fan and Han, 2008*; see *Coll, 2018* for review). Moreover, the mean ERP amplitudes at 140–180 ms predicted self-report of others' pain and one's own unpleasantness (*Fan and Han, 2008*).

Particularly related to the current work are neuroimaging findings that compared brain responses to pain versus neutral expressions. fMRI studies found that viewing video clips (*Botvinick et al., 2005*) or pictures (*Sheng et al., 2014*) showing faces with pain versus neutral expressions or viewing photos of faces of patients who were suffering from provoked pain versus chronic pain (*Saarela et al., 2007*) induced activations in the ACC, AI, and inferior parietal cortex. Moreover, the cortical areas activated by facial expressions of pain were also engaged by the first-hand experience of pain evoked by thermal stimulation (*Botvinick et al., 2005*). Moreover, the strengths of AI activations during observation of others' pain were correlated with subjective feelings of others' pain (*Saarela et al., 2007*). ERP studies found that neural responses to pain expressions occurred as early as 130 ms after face onset over the frontal/central regions as indexed by the increased amplitude of a positive component at 128–188 ms (P2) in response to pain compared neutral expressions (*Sheng and Han, 2012*; *Sheng et al., 2013*; *Sheng et al., 2016*; *Han et al., 2016*; *Li and Han, 2019*). In addition, the P2 amplitudes in response to others' pain expressions positively predicted subjective feelings of own unpleasantness induced by others' pain and self-reports of one's own empathy traits (*Sheng and Han, 2012*). In addition, source estimation of the P2 component in response to others' pain expressions suggested a possible origin in the ACC. Taken together, these brain imaging findings suggest effective subjective and objective measures of empathy (i.e., understanding and sharing of others' pain) that are suitable for the investigation of neural mechanisms underlying modulations of empathy and altruism by BOP.

In Experiment 1, we randomly assigned patient or actor/actress identities to faces to test how experimentally manipulated BOP associated with face identities caused changes in empathy (i.e., subjective evaluation of others' pain) and altruistic behavior (i.e., monetary donations). We predicted that lack of BOP related to actors/actresses (vs. patients) would result in reduced empathy and altruistic behavior. In Experiment 2, based on the common belief that an effective medical treatment reduces a patient's pain, we tested whether decreasing BOP due to knowledge of effective medical treatments of patients also reduced empathy and altruistic behavior.

In Experiments 3 and 4, we investigated whether BOP modulates empathic brain activity by recording EEG signals in response to pain or neutral expressions of faces with patient or actor/actress identities. Brain activities related to empathy were quantified by comparing neural responses to pain versus neutral expressions to exclude neural processes of facial structures, social attributes (e.g., gender), and other empathy-unrelated information. Given previous findings that the P2 amplitude increased to pain compared to neutral expressions and was associated with self-report of sharing of others' pain (*Sheng and Han, 2012*; *Sheng et al., 2013*; *Sheng et al., 2016*; *Han et al., 2016*; *Li and Han, 2019*), we focused on how the P2 amplitude in response to pain (vs. neutral) expressions was modulated by facial identities (i.e., patient or actor/actress) that link to different beliefs (i.e., patients' pain expressions manifest their actual feelings whereas actors/actresses' pain expressions do not). Our ERP results showed evidence that actor/actress compared to patient identities of faces decreased the empathic neural responses (i.e., P2 amplitudes in response to pain [vs. neutral] expressions) within 200 ms post-stimulus. In Experiment 5, we further revealed behavioral and EEG evidence that neural responses to pain expressions of faces mediate BOP effects on empathy and monetary donations.

In Experiment 6, we employed fMRI to examine brain regions in which blood oxygen level-dependent (BOLD) signals are modulated by BOP. We examined BOLD responses to faces that had either patient or actor/actress identities, received painful/non-painful stimulations, and showed pain or neutral expressions. fMRI results allowed us to test whether empathic neural responses in the cognitive (i.e., the dorsal mPFC and TPJ; *Völlm et al., 2006*; *Schnell et al., 2011*; also see *Lamm et al.,*

*2011*; *Fan et al., 2011*; *Shamay-Tsoory, 2011*), sensorimotor/affective (i.e., the ACC, insula, and sensorimotor cortex; *Jackson et al., 2005*; *Singer et al., 2004*; *Avenanti et al., 2005*), or both nodes of the empathic neural network would be modulated by BOP that was manipulated by assigning different identities (i.e., patient or actor/actress) to empathy targets. In addition, we examined whether neural responses in the empathic network would be able to predict variations of subjective feelings of others' pain due to lack of BOP.

Taken together, our behavioral and brain imaging results showed consistent evidence that lack of BOP or decreasing BOP resulted in reduced empathy and altruistic behavior. Our findings suggest that BOP may provide a cognitive basis for human empathy and altruism and uncover intermediate brain mechanisms by which BOP influences empathy and altruistic behavior.

## Results

### Experiment 1: Lack of BOP reduces subjective estimation of empathy and altruistic behavior

In Experiment 1, we tested the predictions that lack of BOP decreases empathy and altruistic behavior by experimentally manipulating individuals' BOP. We presented participants (N=60) with photos of faces of 16 models (half males) with pain expressions (see Materials and methods for details). The participants were informed that these photos were taken from patients who suffered from a disease. In the 1st_round test, the participants were shown with each photo and asked to report the perceived pain intensity of each patient by rating on a Likert-type scale (0=not painful at all, 10=extremely painful). This rating task was adopted from previous research (*Bieri et al., 1990*; *Jackson et al., 2005*; *Lamm et al., 2007*; *Fan and Han, 2008*; *Sheng and Han, 2012*) to assess the participants' understanding of others' pain feeling—a key component of empathy. Thereafter, the participants were invited to donate money to the patient in the photo by selecting an amount from an extra bonus payment for their participation (0–10 points, one point=¥0.2) as a measure of altruistic behavior. The participants were informed that the amount of one of their donation decisions would be selected randomly and endowed to a charity organization to help those who suffered from the same disease. After the 1st_round test, the participants were asked to perform a 5-min calculation task to clean their memory of performances during the 1st_round test. The participants were then informed that this experiment actually tested their ability to recognize facial expressions and the photos were actually taken from eight patients and eight actors/actresses. We expected that identity changes from patients to actors/actresses would decrease BOP because patients' pain expressions reflect their actual emotional states whereas pain expressions performed by actors/actresses do not indicate an actual painful state. The participants were then asked to perform the 2nd_round test in which each photo was presented again with patient or actor/actress identity indicated by a word (i.e., patient, actor, or actress) below the photo. The participants had to perform the same pain intensity rating and donation tasks as those in the 1st_round test. The participants were told that an amount of money would be finally selected from their 2nd_round donation decisions and presented to the same charity organization after the study.

The mean rating scores of pain intensity and amounts of monetary donations were subject to repeated-measures analyses of variance (ANOVAs) of Test Phase (1st_round vs. 2nd_round test)× Identity Change (patient-identity change [patient to actor/actress] vs. patient-identity repetition [patient to patient]) as independent within-subjects variables. As expected, the results revealed that patient-identity change or patient-identity repetition produced opposite effects on both perceived pain intensity and amounts of monetary donations, as indicated by significant interactions of Test Phase×Identity Change (F(1,59)=123.476 and 60.638, ps<0.001, $\eta_p^2$=0.677 and 0.507, 90% confidence interval [CI]=[0.555, 0.747] and [0.351, 0.611], *Figure 1a and b*). Specifically, patient-identity change (i.e., from patients to actors/actresses) significantly reduced perceived pain intensity and amounts of monetary donations in the 2nd_round (vs. 1st_round) test (F(1,59)=82.664 and 34.542, ps<0.001, $\eta_p^2$=0.584 and 0.369, 90% CI=[0.440, 0.673] and [0.207, 0.495]). By contrast, patient-identity repetition significantly increased both perceived pain intensity and monetary donations in the 2nd_round (vs. 1st_round) test (F(1,59)=36.060 and 27.457, ps<0.001, $\eta_p^2$=0.379 and 0.318, 90% CI=[0.216, 0.503] and [0.159, 0.449]). These results suggest that our manipulations of BOP caused reliable changes in subjective evaluation of others' pain and related monetary donations

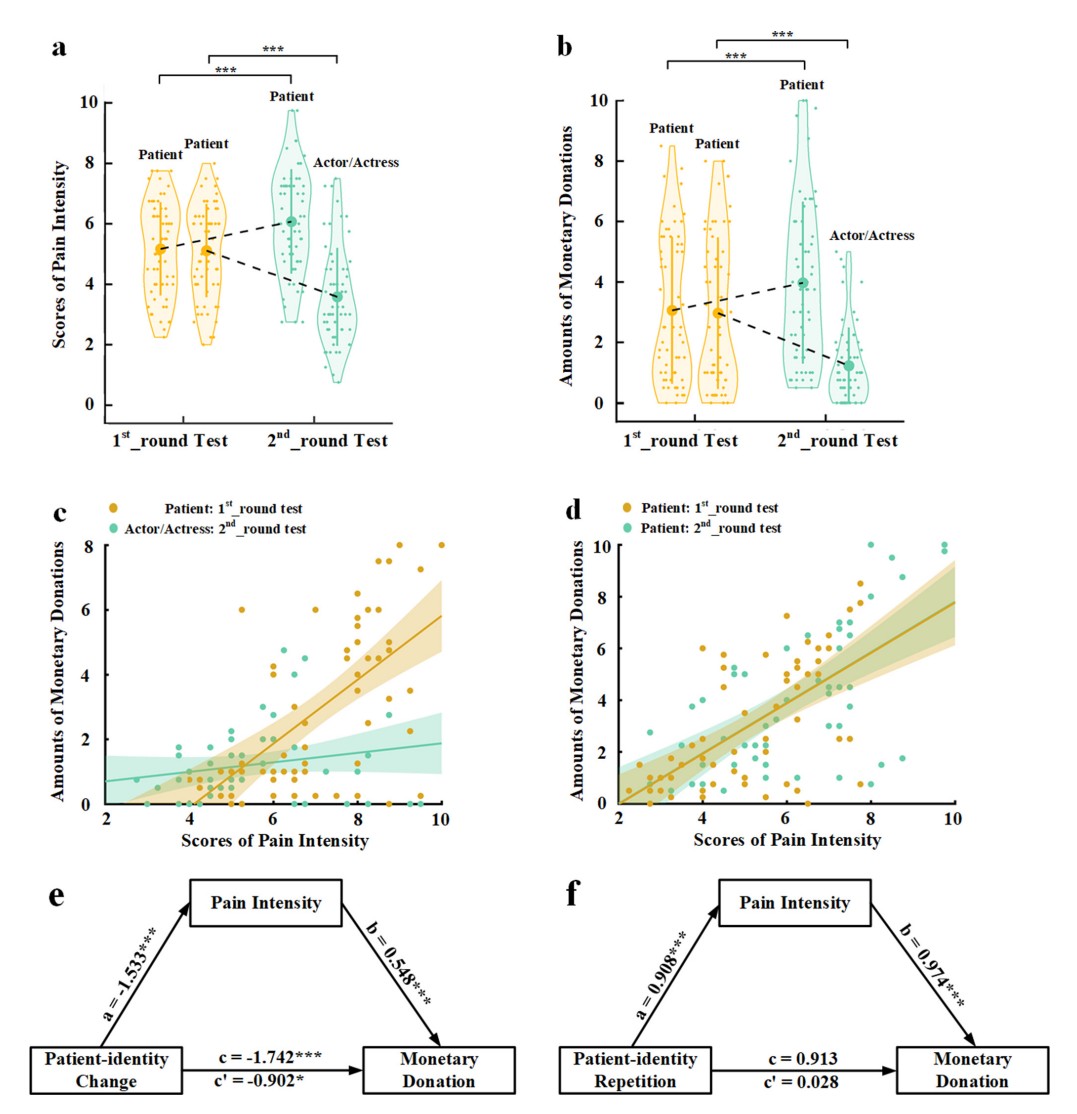

**Figure 1.** Behavioral results in Experiment 1. (a) Mean rating scores of pain intensity in the 1st_round and 2nd_round tests. (b) Mean amounts of monetary donations in the 1st_round and 2nd_round tests. Shown are group means (large dots), standard deviation (bars), measures of each individual participant (small dots), and distribution (violin shape) in (a) and (b). (c) The associations between rating scores of pain intensity and amounts of monetary donations for patients in the 1st_round test and for actors/actresses in the 2nd_round test. (d) The associations between rating scores of pain intensity and amounts of monetary donations for patients in both the 1st_round and 2nd_round tests. (e) Rating scores of pain intensity partially mediate the relationship between patient-identity change and reduced monetary donations. (f) Rating scores of pain intensity mediate the relationship between patient-identity repetition and increased monetary donations.

The online version of this article includes the following source data for figure 1:

**Source data 1.** Pain intensity rating scores.
**Source data 2.** Amounts of monetary donations.

in opposite directions. Interestingly, to some degree rather than not at all, the participants reported pain and donated to faces with actor/actress identity in the 2nd_round test, suggesting that lack of BOP did not fully eliminate empathy and altruistic behavior toward those who showed pain expressions.

To investigate whether perceived pain intensity mediated the relationships between experimentally manipulated BOP and monetary donations, we first conducted Pearson correlation analyses of the relationship between empathy and altruism. The results showed that the rating scores of pain

intensity of faces whose identities changed from patient in the 1st_round test to actor/actress in the 2nd_round test significantly predicted the amount of monetary donations in the 1st_round test but not in the 2nd_round test (r=0.608 and 0.187, p<0.001 and p=0.152, 95% CI=[0.422, 0.776] and [−0.069, 0.435], all results were false discovery rate-corrected, *Figure 1c*). The rating scores of pain intensity also significantly predicted the amount of monetary donations for faces whose patient identities did not change in the 1st_round and 2nd_round tests (r=0.619 and 0.628, ps<0.001, 95% CI= [0.449, 0.776] and [0.417, 0.775], *Figure 1d*). We conducted mediation analyses to further test an intermediate role of empathy between BOP and altruistic behavior (see Materials and methods). The first mediation analysis showed that rating scores of pain intensity partially mediated the relationship between patient-identity change and reduced amount of monetary donations (direct effect: c′=−0.902, t(118)=−2.468, p=0.015, 95% CI=[−1.626, –0.178]; indirect effect: a×b=−0.839, 95% CI=[−1.455, –0.374], *Figure 1e*, see *Supplementary file 1* for statistical details). The second mediation analysis showed evidence that the rating scores of pain intensity also mediated the relationship between patient-identity repetition and increased amount of monetary donations (direct effect: c′=0.028, t(118)=0.072, p=0.943, 95% CI=[−0.727, 0.782]; indirect effect: a×b=0.885, 95% CI= [0.314, 1.563], *Figure 1f*, see *Supplementary file 2* for statistical details). These results indicate a key functional role of BOP in altruistic behavior and suggest changes in subjective evaluation of others' pain as an intermediate mechanism underlying the effect of BOP on monetary donations.

## Experiment 2: Intrinsic BOP predicts subjective estimation of empathy and altruistic behavior

In Experiment 1, BOP was manipulated by randomly assigning patient or actor/actress identities to faces and the results showed that experimentally manipulated BOP changes caused variations of empathy and altruistic behavior. In Experiment 2, we further investigated whether an individual's intrinsic BOP (i.e., various representations of actual emotional states of different faces with pain expressions) can predict empathy and altruistic behavior across different faces. Moreover, as a replication, we tested whether changing the participants' intrinsic BOP causes changes in empathy and altruistic behavior in directions similar to those observed in Experiment 1. In addition, we assessed whether changing intrinsic BOP modulated sharing of others' pain—another key component of empathy (*Bieri et al., 1990*; *Jackson et al., 2005*; *Lamm et al., 2007*; *Fan and Han, 2008*; *Sheng and Han, 2012*). Finally, we tested whether BOP-induced emotional sharing mediates the relationship between BOP and altruistic behavior.

To address these issues, we tested an independent sample (N=60) using the stimuli and procedure that were the same as those in Experiment 1 except the following. In the 1st_round test, the participants were informed that they were to be shown photos with pain expressions taken from patients who suffered from a disease and received a medical treatment. After the presentation of each photo, the participants were asked to estimate, based on perceived pain expression of each face, how effective they believed the medical treatment was for each patient by rating on a Likert-type scale (0=no effect or 0% effective, 100=fully effective or 100% effective). The rating scores were used to estimate the participants' intrinsic BOP of each face with a higher rating score (indicating more effective treatment) corresponding to a weaker BOP because a more effective medical treatment reduces a patient's pain to a greater degree. In addition to rating pain intensity of each face, the participants were asked to report how unpleasant they were feeling when viewing each photo by rating on a Likert-type scale (0=not unpleasant at all, 10=extremely unpleasant). The unpleasantness rating was performed to assess the emotional sharing of others' pain. In the 2nd_round test, the participants were told that the medical treatment was actually fully effective for half patients but had no effect on the others. Each photo was then presented again with information that the medical treatment applied to the patient was 100% effective (to decrease the participants' beliefs of the patients' painful states) or 0% effective (to enhance the participants' beliefs of the patients' painful states). Thereafter, the participants were asked to perform the rating tasks and to make monetary donation decisions, similar to those in the 1st_round test.

To assess whether individuals' intrinsic BOP predicted their empathy and altruistic behavior across different target faces, we conducted Pearson correlation analyses of the relationships between intrinsic BOP as indexed by the rating score of treatment effectiveness and empathy rating scores/ amounts of monetary donations across the 16 models in the 1st_round test in each participant. The correlation coefficients were then transformed to Fisher's z-values that were further compared with

0. One-sample t-tests revealed that the z-values were significantly smaller than 0 (correlations between intrinsic BOP and pain intensity/unpleasantness/monetary donation: mean±s.d. =−0.631±0.531, −0.643±0.524, and −0.469±0.529; t(59)=−9.213, −9.501, and −6.875; ps<0.001; Cohen's d=1.188, 1.227, and 0.887; 95% CI=[−0.768, −0.494], [−0.778, −0.507], and [−0.606, −0.333], *Figure 2a–c*), suggesting that a larger score of treatment effectiveness (i.e., a weaker intrinsic BOP related to a face) predicted weaker empathy and less monetary donations relate to that face. These results provide evidence for associations between intrinsic BOP and empathy/altruism.

Next, we tested whether decreased (or increased) BOP also predicts changes in empathy/altruistic behavior across different target faces for each participant. To do this, we calculated belief changes (decreased BOP: 100% effective minus the participants' initial estimation; enhanced BOP: the participants' initial estimation minus 0% effective), empathy changes (rating scores in the 2nd_round vs. 1st_round test), and changes in altruistic behavior (the amount of monetary donation in the 2nd_round vs. 1st_round test) related to each model in each participant. Similarly, we conducted Pearson correlation analyses to examine associations between changes in beliefs and empathy/donation for decreased BOP patients and enhanced BOP patients, respectively, in each participant. The correlation coefficients were then transformed to Fisher's z-values that were further compared with 0. One-sample t-tests showed that the z-values were significantly smaller than zero for decreased BOP patients (the correlation between changes in belief and pain intensity: z-value [mean±s.d.]=−0.304±0.370, t(59)=−6.352, p<0.001, Cohen's d=0.822, 95% CI=[−0.400, −0.208]; the correlation between changes in belief and unpleasantness: z-value [mean±s.d.]=−0.277±0.455, t(59)=−4.706, p<0.001, Cohen's d=0.609, 95% CI=[−0.394, −0.159]; the correlation between changes in belief and monetary donation: z-value [mean±s.d.]=−0.236±0.410, t(59)=−4.465, p<0.001, Cohen's d=0.576, 95% CI=[−0.342, −0.130]). These results suggest that a greater decrease of BOP related to a face predicted greater reduced empathy and less monetary donations. By contrast, one-sample t-tests showed that the z-values were significantly larger than 0 for enhanced BOP patients (the correlation between changes in belief and pain intensity: z-value [mean±s.d.]=0.286±0.488, t(59)=4.533, p<0.001, Cohen's d=0.586, 95% CI=[0.160, 0.412]; the correlation between changes in belief and unpleasantness: z-value [mean±s.d.]=0.227±0.470, t(59)=3.735, p<0.001, Cohen's d=0.483, 95% CI=[0.105, 0.348]; the correlation between changes in belief and monetary donation: z-value [mean±s.d.]=0.162±0.538, t(59)=2.332, p=0.023, Cohen's d=0.301, 95% CI=[0.023, 0.301]). These results suggest that a greater increase of BOP predicted greater increased empathy and more monetary donations across individual empathy targets. These results provide evidence for associations between changes in BOP and empathy/altruism across different faces for each participant.

To test whether the results in Experiment 2 replicated those in Experiment 1, we conducted ANOVAs of the mean empathy scores and amounts of monetary donations with Test Phase (1st_round vs. 2nd_round) and Belief Change (initial self-rated effectiveness to informed 0% effectiveness vs. initial self-rated effectiveness to informed 100% effectiveness) as independent within-subjects variables. The results showed that decreasing internal BOP (i.e., for 100% effective target faces) resulted in lower subjective evaluation of others' pain and one's own unpleasantness and less monetary donations in the 2nd_round vs. 1st_round test, whereas enhancing BOP (i.e., for 0% effective target faces) produced opposite effects (*Figure 2d–f*, see *Supplementary file 3* for statistical details). These results replicated those in Experiment 1 and provided further evidence that changing BOP resulted in variations of empathy and altruistic behavior.

Pearson correlations analyses of the mean rating scores in the 1st_round and 2nd_round tests across the participants showed that for '100% effective' patients, the 1st_round but not the 2nd_round rating scores of empathy significantly predicted the amount of monetary donations (pain intensity rating: r=0.530 and 0.184, p<0.001 and p=0.159, 95% CI=[0.334, 0.698] and [−0.057, 0.425]; unpleasantness rating: r=0.307 and 0.074, p=0.017 and p=0.576, 95% CI=[0.046, 0.541] and [−0.199, 0.358], *Figure 2g and h*). For '0% effective' patients, however, both the 1st_round and 2nd_round rating scores of empathy significantly predicted the amount of monetary donations (pain intensity rating: r=0.582 and 0.476, ps< 0.001, 95% CI=[0.415, 0.725] and [0.287, 0.638]; unpleasantness rating: r=0.373 and 0.280, p=0.006 and 0.04, 95% CI=[0.096, 0.590] and [0.011, 0.511], *Figure 2i and j*).

Furthermore, the results of mediation analyses showed that rating scores of pain intensity partially mediated the relationship between decreased BOP (i.e., for '100% effective' patients) and monetary donations (direct effect: c'=−0.038, t(58)=−3.657, p<0.001, 95% CI=[−0.059, 0.017]; indirect effect:

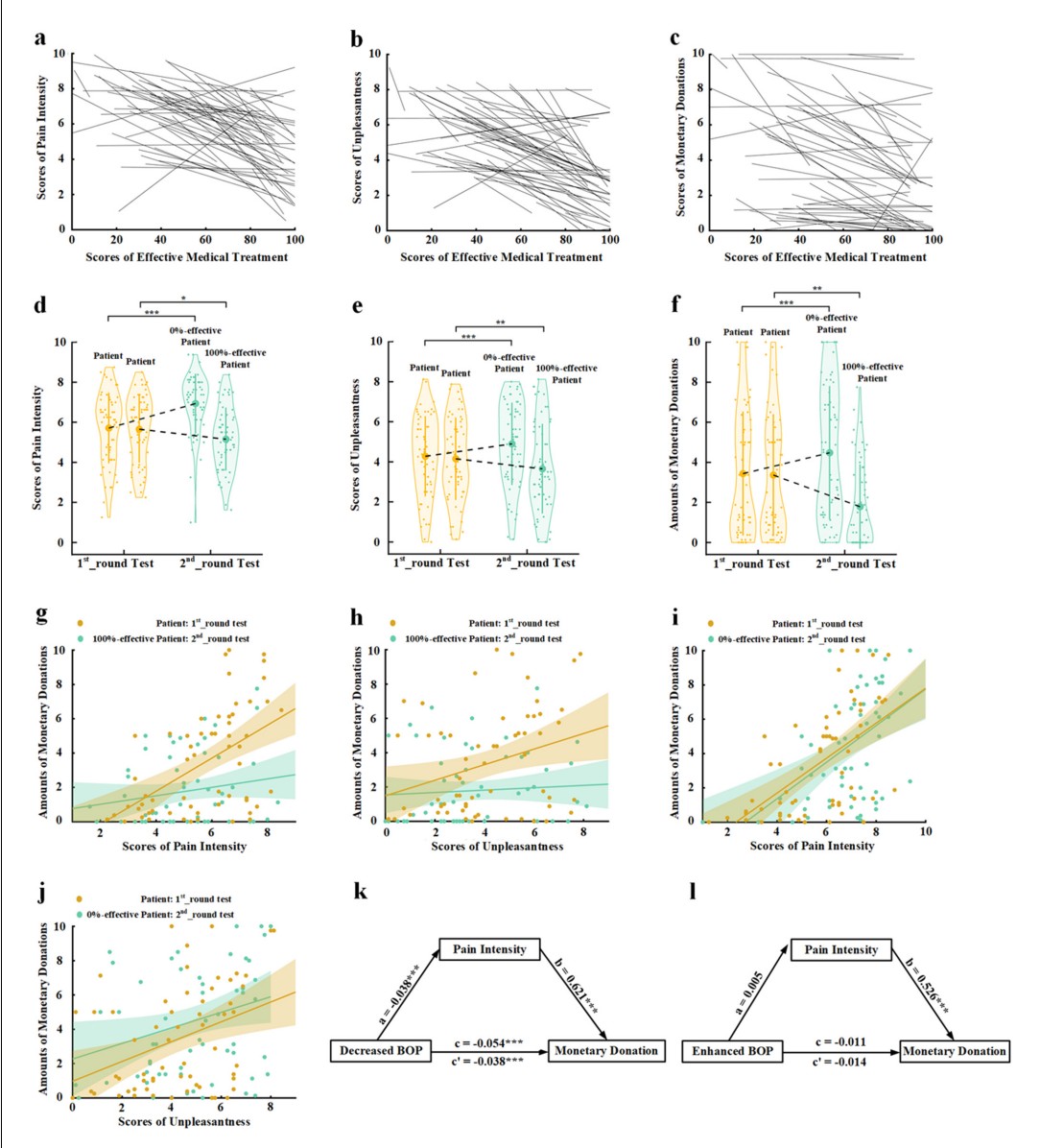

**Figure 2.** Behavioral results in Experiment 2. The relationships between intrinsic BOP (indexed by the rating score of effective medical treatments) and scores of pain intensity (a), own unpleasantness (b), and monetary donations (c), respectively, across the 16 models in the 1st_round test in each participant. The regression line of each participant is plotted in (a–c). (d–f) Mean rating scores of pain intensity, own unpleasantness, and monetary donations in the 1st_round and 2nd_round tests. (g) The associations between rating scores of pain intensity and amounts of monetary donations for patients in the 1st_round test and for 100% effective patients in the 2nd_round test across all the participants. (h) The associations between rating scores of own unpleasantness and amounts of monetary donations for patients in the 1st_round test and for 100% effective patients in the 2nd_round test across all the participants. (i) The associations between rating scores of pain intensity and amounts of monetary donations for patients in the 1st_round test and for 0% effective patients in the 2nd_round test across all the participants. (j) The associations between rating scores of own unpleasantness and amounts of monetary donations for patients in the 1st_round test and for 0% effective patients in the 2nd_round test across all the participants. (k) Rating scores of pain intensity change partially mediate the relationship between decreased BOP and changes in monetary donations. (l) Rating scores of pain intensity change fail to mediate the relationship between enhanced BOP and changes in monetary donations. Shown are group means (large dots), standard deviation (bars), measures of each individual participant (small dots), and distribution (violin shape) in (d–f). BOP, beliefs of others' pain.

The online version of this article includes the following source data for figure 2:

**Source data 1.** Pain intensity rating scores.

**Source data 2.** Own unpleasantness rating scores.

**Source data 3.** Amounts of monetary donations.

a×b=−0.016, 95% CI=[−0.027, –0.005], *Figure 2k*, see *Supplementary file 4* for statistical details). However, rating scores of unpleasantness did not mediate the relationship between decreased BOP and monetary donations (indirect effect: a×b=−0.002, 95% CI=[−0.009, 0.003]). Neither pain intensity nor unpleasantness ratings mediated the relationship between enhanced BOP (i.e., for '0% effective' patients) and monetary donations (indirect effect: a×b=0.003 and −0.002, 95% CI=[−0.009, 0.013] and [−0.007, 0.004], *Figure 2l*, see *Supplementary files 5*, *6*, and *7* for statistical details). These behavioral results suggest that decreased BOP influences altruistic decisions possibly via modulations of the cognitive component of empathy (i.e., understanding others' pain) rather than the affective component of empathy (i.e., sharing others' pain).

## Experiment 3: Lack of BOP decreased empathic brain activity

Experiments 1 and 2 showed evidence that self-report measures of empathy for pain were affected by BOP. In Experiment 3, we further investigated whether and how changing BOP modulates brain activity in response to perceived cues signaling others' pain as an objective estimation of empathy. If BOP provides a basis of empathy of others' pain, lack of BOP should reduce empathic neural responses to visual stimuli signaling others' pain. We tested this assumption by recording EEG to faces of 16 models from an independent sample (N=30). The participants were first presented with these faces with neutral expressions and were informed that these photos were taken from eight patients who suffered from a disease and from eight actors/actresses. The participants were asked to remember the patient or actor/actress identity of each neutral face and had to pass a memory test with a 100% recognition accuracy. Thereafter, the participants were informed that they would be presented with photos of these faces with either neutral or pain expressions, and photos of pain expressions were taken from the patients who were suffering from the disease or from the actors/actresses who imitated patients' pain. The participants were asked to make judgments on the identity of each face (i.e., patient vs. actor/actress) with a neutral or pain expression by pressing one of two buttons while EEG was recorded. After EEG recording, the participants were asked to rate pain intensity of each face with pain or neutral expression on a Likert-type scale (0=not painful at all, 7=extremely painful) and to what degree they believed in the identity of each face with pain expression on a 15-point Likert-type scale (−7=extremely believed as an actor/actress, 0=not sure, 7=extremely believed as a patient). Because the same set of stimuli were perceived as patients or actors/actresses across the participants, modulations of brain activity in response to pain expressions only reflected the effects of BOP concomitant with the face identity (i.e., real pain for patients but fake pain for actors/actresses).

The participants reported a positive mean belief score corresponding to faces with a patient identity (2.496±2.51) but a negative mean belief score corresponding to faces with an actor/actress identity (−2.210±3.25) (t(29)=4.932, p<0.001, Cohen's d=0.900, 95% CI=[2.755, 6.658]), suggesting successes of our manipulations of face identities. An ANOVA of the mean rating scores of pain intensity with Identity (patient vs. actor/actress) and Expression (pain vs. neutral) as within-subject variables revealed a significant Identity×Expression interaction (F(1,29)=4.905, p=0.035, $\eta_p^2$=0.145, 90% CI=[0.006, 0.330], *Figure 3a*), suggesting greater subjective feelings of pain intensity for faces with patient compared to actor/actress identity. Moreover, a larger score of belief of patient identities significantly predicted greater subjective feelings of pain intensity related to patients' pain (vs. neutral) expressions (r=0.384, p=0.036, 95% CI=[0.074, 0.627]), whereas there was no significant association between belief scores and subjective feelings of pain intensity related to actors/actresses' pain (vs. neutral) expressions (r=0.264, p=0.159, 95% CI=[−0.162, 0.605]). These results provide further evidence for a link between BOP and empathy for patients' pain.

The participants responded to face identities with high accuracies during EEG recording (>81% across all conditions, see *Supplementary file 8* for details). ERPs to face stimuli in Experiment 3 were characterized by an early negative activity at 95–115 ms (N1) and a positive activity at 175–195 ms (P2) at the frontal/central regions, which were followed by two positive activities at 280–340 ms (P310) over the parietal region and 500–700 ms (P570) over the frontal area (*Figure 3b*). Previous ERP studies have shown that empathic neural responses to pain expressions are characterized by an increased P2 amplitude and the P2 amplitude to pain (vs. neutral) expressions predicts self-report of affective sharing (*Sheng and Han, 2012*; *Sheng et al., 2016*; *Luo et al., 2018*; *Li and Han, 2019*). Therefore, our ERP data analyses focused on whether BOP modulates the P2 amplitude to pain (vs. neutral) expressions given the previous ERP findings. ANOVAs of the P2 amplitudes with Identity

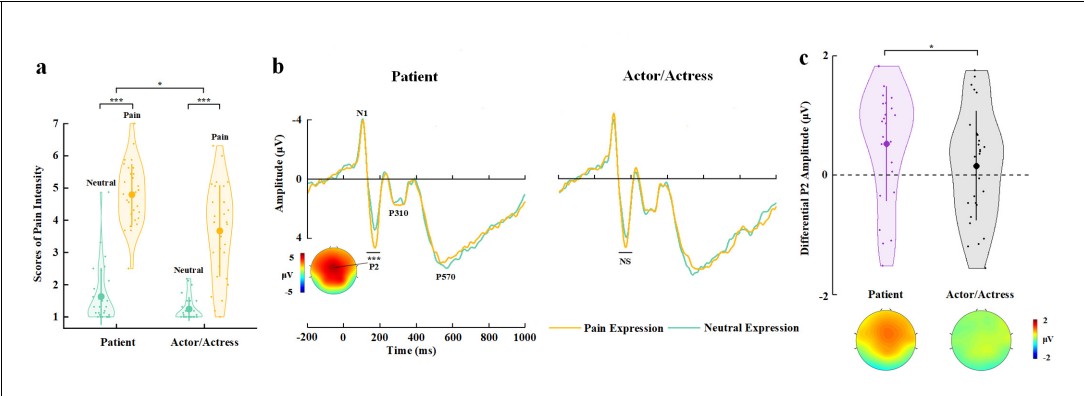

**Figure 3.** EEG results of Experiment 3. (**a**) Mean rating scores of pain intensity to pain versus neutral expressions of faces with patient or actor/actress identities. (**b**) ERPs to faces with patient or actor/actress identities at frontal electrodes. The voltage topography shows the scalp distribution of the P2 amplitude with the maximum over the central/frontal region. (**c**) Mean differential P2 amplitudes to pain versus neutral expressions of faces with patient or actor/actress identities. The voltage topographies illustrate the scalp distribution of the P2 difference waves to pain versus neutral expressions of faces with patient or actor/actress identities, respectively. Shown are group means (large dots), standard deviation (bars), measures of each individual participant (small dots), and distribution (violin shape) in (**a**) and (**c**). EEG, electroencephalography; ERP, event-related potential.

The online version of this article includes the following source data for figure 3:

**Source data 1.** Pain intensity rating scores.
**Source data 2.** Mean differential P2 amplitudes.

(patient vs. actor/actress) and Expression (pain vs. neutral) as within-subject variables revealed a significant Identity×Expression interaction ($F(1,29)=7.490$, $p=0.010$, $\eta_p^2=0.205$, 90% CI=[0.029, 0.391], see *Supplementary file 9* for statistical details). Simple effect analyses verified significantly greater P2 amplitudes to pain versus neutral expressions of patients' faces ($F(1,29)=18.059$, $p<0.001$, $\eta_p^2=0.384$, 90% CI=[0.150, 0.546]), whereas the P2 amplitude did not differ significantly between pain and neutral expressions of actors/actresses' faces ($F(1,29)=0.334$, $p=0.568$, $\eta_p^2=0.011$, 90% CI= [0.000, 0.135], *Figure 3b and c*). We further conducted Bayes factor analyses to examine the null effect of pain expressions on the P2 amplitudes to actors/actresses' faces. The Bayes factor represents the ratio of the likelihood of the data fitting under the alternative hypothesis versus the likelihood of fitting under the null hypothesis. The results showed a Bayes factor of 0.227 which provided further evidence for the null hypothesis. The results indicate that, while the effect of pain (vs. neutral) expression on the P2 amplitudes to patients' faces was similar to our previous findings that the P2 amplitudes increased to pain (vs. neutral) expressions of face without patient identities (*Sheng and Han, 2012*; *Sheng et al., 2016*), the P2 amplitude was less sensitive to pain versus neutral expressions of faces with actor/actress identities. This finding indicates that lack of BOP significantly weakens early empathic neural responses to others' pain within 200 ms after stimulus onset.

## Experiment 4: BOP is necessary for modulations of empathic brain activity

The learning and EEG recording procedures in Experiment 3 consisted of multiple processes, including learning, memory, and recognition of face identities, assignment to different social groups (e.g., patient or actor groups), and so on. The results of Experiment 3 left an open question of whether these processes, even without BOP changes induced through these processes, would be sufficient to result in modulations of the P2 amplitude in response to pain (vs. neutral) expressions of faces with different identities. In Experiment 4, we addressed this issue using the same learning and identity recognition procedures as those in Experiment 3 except that the participants in Experiment 4 had to learn and recognize the identities of faces of two baseball teams and that there is no prior difference in BOP associated with individual faces from the two baseball teams. If the processes involved in the learning and reorganization procedures rather than the difference in BOP were sufficient for modulations of the P2 amplitude in response to pain (vs. neutral) expressions of faces, we would expect similar P2 modulations in Experiments 3 and 4. Otherwise, if the difference in BOP

produced during the learning procedure was necessary for the modulation of empathic neural responses, we would not expect modulations of the P2 amplitude in response to pain (vs. neutral) expressions in Experiment 4.

We clarified these predictions in an independent sample (N=30) in Experiment 4. We employed the stimuli and procedure that were the same as those in Experiment 3 except that, during the learning phase, the participants were informed that the 16 models were from two baseball teams (half from Tiger team and half from Lion team) and they suffered from a disease. After the participants had remembered the team identity of each neutral face in a procedure similar to that in Experiment 3, they performed identity (i.e., Tiger vs. Lion team) judgments on the faces with neutral or pain expressions during EEG recording. This manipulation built team identities should not influence self-report and EEG estimation of empathy because the Tiger/Lion team identities did not bring any difference in BOP between pain expressions of faces from the two teams.

The participants responded to face identities with high accuracies during EEG recording (>79% across all conditions). Rating scores of pain intensity did not differ significantly between faces from the two teams (F(1,29)=1.608, p=0.215, $\eta_p^2$=0.053, 90% CI=[0, 0.216], Bayes factors=0.261, *Figure 4a*, see *Supplementary file 10* for details). ANOVAs of the mean P2 amplitudes over the frontal electrodes revealed a significant main effect of facial expression (F(1,29)=12.182, p=0.002, $\eta_p^2$=0.296, 90% CI=[0.081, 0.473], *Figure 4b and c*, see *Supplementary file 11* for details), as the P2 amplitude was enlarged by pain compared to neutral expressions. However, this effect did not differ significantly between faces from the two teams (F(1,29)=0.040, p=0.843, $\eta_p^2$=0.001, 90% CI=[0, 0.053], Bayes factors=0.258). The null interaction effect on either self-report of empathy and the P2 amplitudes to pain (vs. neutral) expressions in Experiment 4 was not simply due to an underpowered sample size because the same sample size in Experiment 3 revealed reliable BOP effects on self-report and EEG (i.e., the P2 amplitude) estimation of empathy. Taken together, the results in Experiments 3 and 4 suggest a key role of BOP, but not other cognitive processes involved in the experimental manipulations, in modulations of neural responses to others' pain.

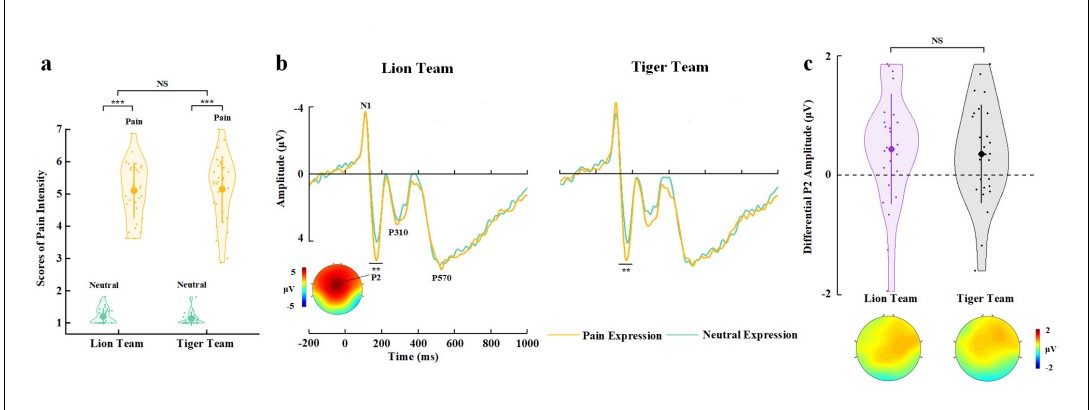

**Figure 4.** EEG results of Experiment 4. (a) Mean rating scores of pain intensity to pain versus neutral expressions of faces with Lion or Tiger team identities. (b) ERPs to faces with Lion/Tiger team identities at frontal electrodes. The voltage topography shows the scalp distribution of the P2 amplitude with the maximum over the central/frontal region. (c) Mean differential P2 amplitudes to pain versus neutral expressions of faces with Lion/Tiger team identities. The voltage topographies illustrate the scalp distribution of the P2 difference waves to pain versus neutral expressions of faces with the Lion/Tiger team identities, respectively. Shown are group means (large dots), standard deviation (bars), measures of each individual participant (small dots), and distribution (violin shape) in (a) and (c). EEG, electroencephalography; ERP, event-related potential.

The online version of this article includes the following source data for figure 4:

**Source data 1.** Pain intensity rating scores.
**Source data 2.** Mean differential P2 amplitudes.

## Experiment 5: Empathic brain activity mediates relationships between BOP and empathy/altruistic behavior

Given that Experiments 1–4 showed consistent evidence for BOP effects on subjective feelings of others' pain, altruistic behavior, and empathic neural responses, in Experiment 5, we further examined whether BOP-induced changes in empathic brain activity plays a mediator role in the pathway from belief changes to altered subjective feelings of others' pain and altruistic decisions. To this end, we conducted two-session tests of an independent sample (N=30). In the first session, we employed the stimuli and procedure that were identical to those in Experiment 1 to assess BOP effects on empathy and altruistic behavior. In the second session, we recorded EEG from the participants using the same stimuli and procedure as those in Experiment 3 to examine BOP effects on empathic neural responses. BOP-induced changes in empathic brain activity, rating scores of pain intensity, and amounts of monetary donations recorded in the two-session tests were then subject to mediation analyses.

To assure the participants' beliefs about patient and actor/actress identities of perceived faces, after EEG recording, we asked the participants to complete an implicit association test (IAT) (*Greenwald et al., 1998*) that measured reaction times to faces with patient and actor/actress identities and words related to patients and actors/actresses (see Materials and methods). The D score was then calculated based on response times (*Greenwald et al., 2003*) to assess implicit associations between patient and actor/actress faces and the relevant words. One-sample t-test revealed that the D score was significantly larger than 0 (0.929±0.418, t(29)=12.178, p<0.001, Cohen's d=2.223, 95% CI=[0.773, 1.085]), suggesting that patient faces were more strongly associated with patient relevant words whereas actor/actress faces were more strongly associated with actor/actress relevant words. The results indicate successful belief manipulations during the two-session tests.

The behavioral results in the first-session test replicated the findings of Experiment 1. In particular, decreasing BOP (i.e., changing patient identity in the 1st_round test to actor/actress identity in the 2nd_round test) significantly reduced self-report of others' pain and monetary donations (Test Phase×Identity Change) interactions on rating scores of pain intensity and amounts of monetary donations: (F(1,29)=59.654 and 129.696, ps<0.001, $\eta_p^2$=0.673 and 0.817, 90% CI=[0.479, 0.764] and [0.694, 0.868]); effects of patient-to-actor/actress identity change on rating scores of pain intensity and amounts of monetary donations: (F(1,29)=58.196 and 180.022, ps<0.001, $\eta_p^2$=0.667 and 0.861, 90% CI=[0.472, 0.760] and [0.765, 0.900], *Figure 5a and b*). However, patient-identity repetition failed to significantly increase rating scores of pain intensity and amounts of monetary donations (F(1,29)=0.016 and 0.209, p=0.901 and 0.651, $\eta_p^2$=0.001 and 0.007, 90% CI=[0, 0.022] and [0, 0.119]), possibly due to ceiling effects of our measures in the participants (i.e., larger mean rating scores of pain intensity and mean amounts of monetary donations in the 1st_round test in Experiment 5 than in Experiment 1).

The participants responded to face identities with high accuracies during EEG recording (>83% across all conditions). The EEG results replicated those in Experiment 3 by showing significantly deceased P2 amplitudes to pain (vs. neutral) expressions of actor/actress compared to patient faces (Identity×Expression interaction: F(1,29)=9.494, p=0.004, $\eta_p^2$=0.247, 90% CI=[0.050, 0.429], *Figure 5c and d*, see *Supplementary file 12* for statistical details). Simple effect analyses verified significantly greater P2 amplitudes to pain (vs. neutral) expressions for patients' faces (F(1,29)=17.409, p<0.001, $\eta_p^2$=0.375, 90% CI=[0.142, 0.539]) but not for faces of actors/actresses (F(1,29)=0.270, p=0.607, $\eta_p^2$=0.009, 90% CI=[0, 0.127], Bayes factor=0.220). These behavioral and EEG results are consistent with those in Experiments 1 and 3 and provide repeated evidence for BOP effects on subjective feelings of others' pain, altruistic behavior, and empathic brain activity in the same sample.

Next, we tested a serial mediation model of the relationship between decreased BOP (i.e., identity change from patient to actor/actress) and changes in monetary donations with two mediator variables including empathic neural responses (as indexed by the differential P2 amplitude to pain vs. neutral expressions) and changes in subjective feelings of others' pain (as indexed by differential rating scores of pain intensity) (see Materials and methods for details). This model includes three paths: (1) the indirect effect of patient-identity change on monetary donation via the P2 amplitude ($a_1 \times b_1$=0.219, 95% CI=[−0.141, 0.745]); (2) the indirect effect of patient-identity change on monetary donation via pain intensity ($a_2 \times b_2$=−1.182, 95% CI=[−2.048, −0.510]); and (3) the indirect effect

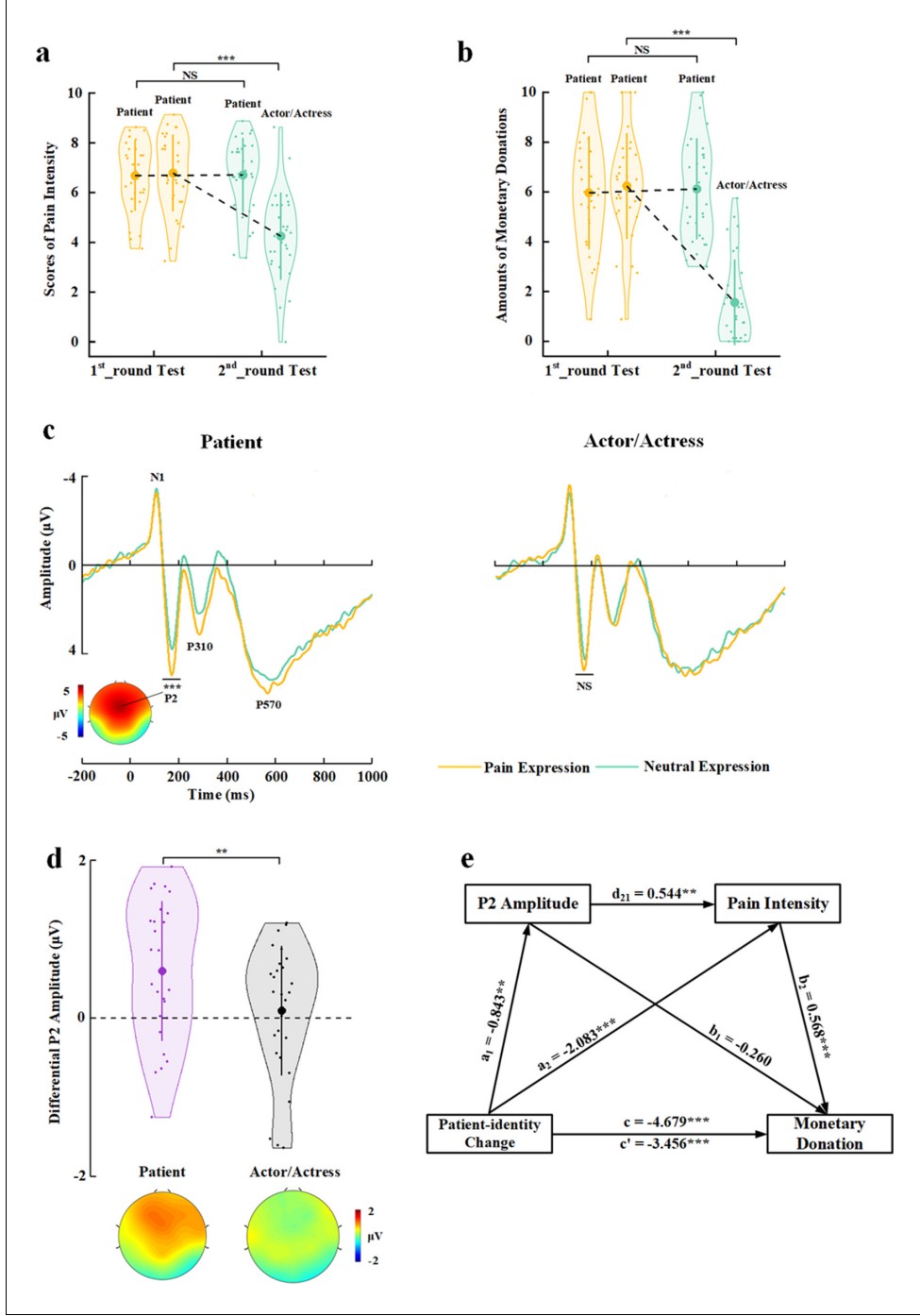

**Figure 5.** Behavioral and EEG results of Experiment 5. (**a**) Mean rating scores of pain intensity in the 1st_round and 2nd_round tests. (**b**) Mean amounts of monetary donations in the 1st_round and 2nd_round tests. (**c**) ERPs to faces with patient or actor/actress identities at frontal electrodes. The voltage topography shows the scalp distribution of the P2 amplitude with the maximum over the central/frontal region. (**d**) Mean differential P2 amplitudes to pain versus neutral expressions of faces with patient or actor/actress identities. The voltage topographies illustrate the scalp distribution of the P2 difference waves to pain versus neutral expressions of faces with patient or actor/actress identities, respectively. (**e**) Illustration of the serial mediation model of the relationship between decreased BOP and changes in monetary donations. Shown are group means (large dots), standard deviation (bars), measures of each individual participant (small dots), and distribution (violin shape) in (**a**), (**b**), and (**d**). BOP, beliefs of others' pain; EEG, electroencephalography; ERP, event-related potential.

The online version of this article includes the following source data for figure 5:

**Source data 1.** Pain intensity rating scores.

**Source data 2.** Amounts of monetary donations.
**Source data 3.** Mean differential P2 amplitudes.

of patient-identity change on monetary donation via P2 amplitude×pain intensity ($a_1 \times d_{21} \times b_2 = -0.261$, 95% CI=[−0.584, –0.059], *Figure 5e*, see *Supplementary file 13* for statistical details). The total indirect effect of patient-identity change on the monetary donation after controlling all indirect effect was $c' = -1.223$, 95% CI=(−2.145, –0.400), which explained 26.14% variance of the total effect of patient-identity change on monetary donation. The effect sizes of the indirect path (2) and (3) were 25.26% and 5.58%, respectively, indicating that subjective feelings of others' pain mediated the association between patient-identity change and reduced monetary donations. Moreover, this mediator role was partially mediated by BOP-induced variations of empathic brain activity in response to others' pain expressions. Taken together, the results of these mediation analyses suggest a pathway from changes in BOP to varied empathic brain activity and changes in subjective report of empathy for other's pain (i.e., the degree of perceived pain in others), which further accounted for BOP-induced changes in monetary donations.

## Experiment 6: Neural structures underlying BOP effects on empathy

While our EEG results revealed evidence for modulations of empathic neural responses by BOP, neural structures underlying these modulation effects remain unclear. In particular, it is unknown whether brain responses underlying cognitive and affective components of empathy are similarly sensitive to the influence of BOP. Therefore, in Experiment 6, we used fMRI to record BOLD signals from an independent sample (N=31) to examine neural architectures in which empathic activities are modulated by BOP. Similarly, the participants were first shown with photos of neutral faces of 20 models and had to remember their patient (10 models) or actor/actress (10 models) identities. After the participants had performed 100% correct in a memory task to recognize the models' identities, they were scanned using fMRI when viewing video clips of the models whose faces received painful (needle penetration) stimulation and showed pain expressions or received non-painful (cotton swab touch) stimulation and showed neutral expressions, similar to those used in the previous studies (*Han et al., 2009*; *Luo et al., 2014*; *Han et al., 2017*). Before scanning the participants were informed that these video clips were recorded from 10 patients who were receiving medical treatment and 10 actors/actresses who practiced to imitate patients' pain expressions. The participants responded to face identity (patient vs. actor/actress) of each model after viewing each video clip by pressing one of two buttons with high accuracies (>80% across all conditions, see *Supplementary file 14* for details).

After fMRI scanning, the participants were presented with each video clip again and had to rate the model's pain intensity and their own unpleasantness. The participants were also asked to rate the degree to which they believed in the models' patient or actor/actress identities in painful video clips on a 15-point Likert-type scale (−7=extremely believed as an actor/actress, 0=not sure, 7=extremely believed as a patient) (see Materials and method, *Supplementary file 14* for results). The mean rating scores confirmed significant differences in beliefs of patient and actors/actresses identities (2.776±3.20 vs. −4.890±1.44, t(30)=10.526, p<0.001, Cohen's d=1.890, 95% CI=[6.178, 9.153]), indicating successful identity manipulations.

We first localized empathic neural responses by conducting a whole-brain analysis of BOLD responses to perceived painful versus non-painful stimuli applied to targets (collapsed faces with patient and actor/actress identities). This analysis revealed significant activations in the cognitive, affective, and sensorimotor nodes of the empathy network, including the bilateral AI/inferior frontal cortex (Montreal Neurological Institute (MNI) peak coordinates x/y/z=−45/17/–5 and 45/26/–8), bilateral inferior and superior temporal gyri (−48/–70/–2 and 51/–58/–5), mPFC (3/56/25), left inferior parietal lobe (−63/–25/31), right superior parietal lobe (30/−58/55), and right post-central gyrus and posterior insula (58/−25/26, *Figure 6a*; all activations were identified using a combined threshold of voxel-level p< 0.001, uncorrected, and cluster-level p<0.05, Family-wise error (FWE)-corrected). These brain activations are similar to those observed in previous research (e.g., *Luo et al., 2014*). To examine brain activity engaged in representing facial identities independent of perceived painful stimulation and pain expressions, we conducted a whole-brain analysis of the contrast of the

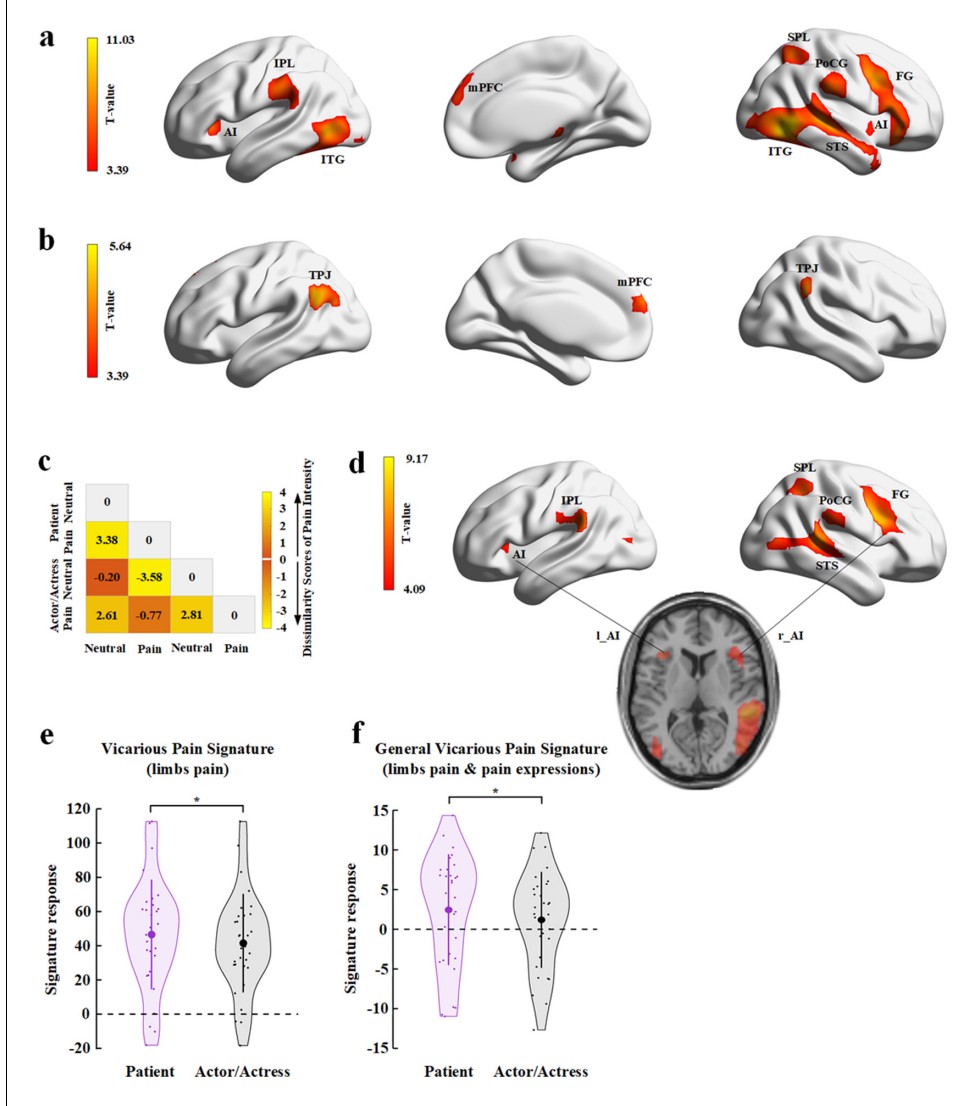

**Figure 6.** fMRI results of Experiment 6. (**a**) Brain activations in response to perceived painful (vs. non-painful) stimuli applied to targets (collapsed faces with patient and actor/actress identities). (**b**) Brain activations in response to non-painful stimuli to patients compared to actors/actresses. (**c**) Illustration of the behavioral dissimilarity matrix (DM) derived from the rating scores of pain intensity across all participants. Each cell in the DM represents the mean difference in rating scores of pain intensity between each pair of conditions. (**d**) Brain activations that were correlated with the behavioral DM revealed in the searchlight RSA. (**e**) Illustration of the vicarious pain signature (defined by response to perceived noxious stimulation of body limbs) responses to patients' and to actors/actresses' pain. (**f**) Illustration of the general vicarious signature (defined by response to perceived noxious stimulation of body limbs and painful facial expressions) responses to patients' and actors/actresses' pain. AI, anterior insula; FG, frontal gyrus; IPL, inferior parietal lobe; ITG, inferior temporal gyrus; MFC, middle frontal cortex; mPFC, medial prefrontal cortex; PoCG, post-central gyrus; RSA, representational similarity analysis; SPL, superior parietal lobe; STS, superior temporal sulcus; TPJ, temporoparietal junction.

The online version of this article includes the following source data for figure 6:

**Source data 1.** Brain activations in response to painful vs. non-painful stimuli (collapsed faces with patient and actor/actress identities).
**Source data 2.** Brain activations in response to non-painful stimuli to patients compared to actors/actresses.
**Source data 3.** Behavioral dissimilarity matrix derived from the rating scores of pain intensity across all participants.
**Source data 4.** Brain activations that were correlated with the behavioral dissimilarity matrix revealed in the searchlight RSA.
**Source data 5.** Data of the vicarious pain signature.
**Source data 6.** Data of the general vicarious pain signature.

stimuli showing non-painful stimulations to patient versus actor/actress. This analysis showed significant activations in the mPFC (−6/59/25) and bilateral TPJ (−54/−58/28 and 57/−67/31, *Figure 6b*; all activations were identified using a combined threshold of voxel-level p<0.001, uncorrected, and cluster-level p<0.05, FWE-corrected).

We conducted a whole-brain univariate analysis to examine the interaction effect (patient vs. actor×pain vs. neutral) on brain activities in response to video clips but did not find a significant effect. Therefore, we further conducted multivariate analyses of BOLD signals to assess neural correlates of BOP effects on subjective feeling of others' pain. Specifically, we conducted a representational similarity analysis (RSA) (*Nili et al., 2014*) of brain activity using a dissimilarity matrix (DM) constructed from scores of pain intensity in different conditions. The RSA sought to find patterns of brain activities in the empathy neural network which can predict the pattern of subjective feeling of others' pain that varied due to BOP. To do this, we first conducted ANOVAs of the mean rating scores and found a significant Identity (patient vs. actor/actress)×Expression (pain vs. neutral) interaction on the rating scores of pain intensity (F(1,30)=5.370, p=0.027, $\eta_p^2$=0.152, 90% CI=[0.029, 0.391]) but not on the rating scores of unpleasantness (F(1,30)=3.945, p=0.056, $\eta_p^2$=0.116, 90% CI= [0, 0.296], see *Supplementary file 14* for statistical details). Simple effect analyses showed significantly larger scores of pain intensity for pain expressions of patients (vs. actors/actresses) (F(1,30) =9.823, p=0.004, $\eta_p^2$=0.247, 90% CI=[0.053, 0.427]), whereas scores of pain intensity did not differ significantly between neutral faces with patient and actor/actress identifies (F(1,30)=2.829, p=0.103, $\eta_p^2$=0.086, 90% CI=[0, 0.260]). The results suggested a clear boundary between subjective feelings of pain intensity in different conditions. Thus we constructed a 4×4 DM for each participant with each cell in the DM representing the mean difference in rating scores of pain intensity between each pair of conditions, as illustrated in *Figure 6c*.

Next, we conducted a searchlight RSA to identify brain regions in which the pairwise similarity of neural responses in the four conditions (two Expressions×two Identities) corresponded to the behavioral DM in each participant (see Materials and methods for details). We first conducted a whole-brain searchlight RSA for each participant. The searchlight results of all participants were then subject to a second group-level analysis to examine the voxels in the empathy network, defined based on the results of the whole-brain contrast of painful versus non-painful stimuli applied to targets, that passed a threshold of voxel-level p<0.05, FWE-corrected. The results revealed significant activations in the left AI (MNI peak coordinates x/y/z=−39/20/8) and inferior parietal cortex (−60/−19/29), and the right AI/frontal cortex (36/23/11), superior temporal gyrus (54/−37/11), inferior post-central gyrus (63/−40/26), and superior parietal cortex (39/−49/50) (*Figure 6d*).

Finally, we estimated BOP effects on neural responses in a vicarious pain signature (VPS) map that was identified to be sensitive to perceived painful stimulations applied to others but not to self-experienced pain (*Krishnan et al., 2016*). We calculated the VPS pattern responses to video clips showing patient or actor/actress faces that received painful (needle penetration) or non-painful (cotton swab touch) stimulation using both the body-specific VPS map in response to perceived noxious stimulation of body limbs (*Krishnan et al., 2016*) and the general VPS in response to both perceived noxious stimulation of body limbs and painful facial expressions (*Zhou et al., 2020a*). We tested the hypothesis of decreased VPS responses to actors/actresses' compared to patients' pain (i.e., lack of BOP reduces empathic brain activities) by conducting t-tests of BOLD signals in VPS maps. The results showed that activities in the VPS pattern were significantly decreased in response to video clips showing actors/actresses' compared to patients' pain (*Figure 6e and f*, body-specific VPS: mean±s.d.=41.487±28.794 vs. 46.548±32.051, t(30)=−2.059, $p_{(one-tailed)}$=0.024, $BF_{+0}$=2.361; general VPS: mean±s.d.=1.188±6.058 vs. 2.462±6.997, t(30)=−2.447, $p_{(one-tailed)}$=0.010, $BF_{+0}$=4.820). These results provide further evidence for decreased empathic brain activities due to lack of BOP for actors/actresses' pain in the empathic neural network.

## Discussion

We conducted six experiments to investigate psychological and neural mechanisms underlying BOP impacts on empathy and altruistic behavior in humans. We manipulated individuals' BOP by randomly assigning patient or actor/actress identities to faces as there was a lack of BOP for actors/actresses' faces but not for patients' faces. We also estimated individuals' intrinsic BOP by asking the participants to estimate the effectiveness of medical treatments of patients to trigger BOP as an

effective medical treatment reduces a patient's pain. We further measured brain activity using EEG and fMRI to examine BOP effects on empathic neural responses with high temporal and spatial resolutions, respectively. Our behavioral and neuroimaging findings showed evidence for a functional role of BOP in modulations of the perception-emotion-behavior reactivity by illustrating how BOP predicted and affected self-reports of empathy, empathic brain activities, and monetary donations. Our findings suggest that BOP may provide a cognitive basis for empathy and altruistic behavior in humans.

Experiments 1 and 2 showed behavioral evidence that manipulated changes in BOP caused subsequent variations of self-report of empathy and altruistic behavior along the directions as predicted. Specifically, decreasing BOP concomitant with changes in face identities (from patient to actor/ actress) or changes in effective medical treatments (from suffering due to a disease to recovery due to medical treatment) significantly reduced self-report of both cognitive (perceived intensity of others' pain) and affective (own unpleasantness induced by perceived pain in others) components of empathy. Decreasing BOP also inhibited following altruistic behavior that was quantified by the amount of monetary donations to those who showed pain expressions. By contrast, reassuring patient identities in Experiment 1 or by noting the failure of medical treatment related to target faces in Experiment 2 increased subjective feelings of others' pain and own unpleasantness and prompted more monetary donations to target faces. The increased monetary donations might be due to repeatedly confirming patient identity or knowing the failure of medical treatment increased the belief of authenticity of targets' pain and thus enhanced cognitive and affective components of empathy. Alternatively, repeatedly confirming patient identity or knowing the failure of medical treatment might activate other emotional responses to target faces such as pity or helplessness, which might also influence altruistic decisions. The increased empathy rating scores and monetary donations might also reflect a contrast effect due to rating patient and actor/actress targets alternately. These possible accounts can be clarified in future work by asking participants to report their emotions and performing rating tasks on patient and actor/actress targets in separate blocks of trials. In consistent with the effects of manipulated BOP on empathy and altruism across the participants, the results of Experiment 2 showed that individuals' intrinsic BOP related to each target face predicted their self-report of empathy and altruistic behavior across different target faces. Moreover, decreased (or increased) intrinsic BOP also predicted changes in empathy/altruistic behavior across different target faces. These converging behavioral findings across different participants and across different target faces provide evidence for causal relationships between BOP and empathy/altruism.

Our results showed that self-reports of others' pain intensity and own unpleasantness elicited by perception of others' pain were able to positively predict altruistic behavior across individuals. Previous research using questionnaire measures of empathy ability found that empathy as a trait is positively correlated with the amount of money shared with others in economic games (*Edele et al., 2013*; *Li et al., 2019*). Taken together, these findings are consistent with the proposition that empathy, as either an instant emotional response to others' suffering (e.g., estimated in our study) or a personality trait (e.g., estimated in *Edele et al., 2013* and *Li et al., 2019*), plays a key role in driving altruistic behavior (*Batson, 1987*; *Batson et al., 2015*; *Eisenberg et al., 2010*; *Hofman, 2008*; *Penner et al., 2005*). Our mediation analyses of the behavioral data in both Experiments 1 and 2 further revealed that the effects of decreased BOP on monetary donations were mediated by self-report of others' pain intensity. These results further suggest empathy as an intermediate mechanism of the BOP effects on altruistic behavior.

Our neuroimaging experiments went beyond the subjective estimation of the relationships between BOP and empathy/altruism by investigating neural mechanisms underlying BOP effects on empathy for others' pain. It is necessary to conduct the objective estimation of empathy to examine BOP effects because self-report measures of empathy can be influenced by social contexts and are unable to unravel brain mechanisms underlying BOP effects on empathy (e.g., *Sheng and Han, 2012*). Our EEG results in Experiments 3 and 5 repeatedly showed that neural responses to pain (vs. neutral) expressions over the frontal regions within 200 ms after face onset (indexed by the P2 amplitude over the frontal/central electrodes) were significantly reduced to faces with actor/actress identities compared to those with patient identities. The results in Experiments 3 and 4 indicate that BOP concomitant with face identity (i.e., patients' pain expressions manifest their actual painful emotional states whereas actors/actresses' pain expressions do not) rather than face identity (e.g., Tiger or Lion team identities) alone resulted in modulations of the P2 amplitudes to pain expressions in the

direction as expected. Numerous EEG studies have shown that the frontal P2 component responds with enlarged amplitudes to various facial expressions such as fear, anger, happy (*Williams et al., 2006*; *Luo et al., 2010*; *Calvo et al., 2013*), and pain (*Sheng and Han, 2012*; *Sheng et al., 2013*; *Sheng et al., 2016*) expressions compared to neutral faces. These findings uncovered early affective processing by differentiating emotional and neutral expressions. ERPs to others' pain within 200 ms post-stimulus occur regardless of task demands and are associated with spontaneous empathy for pain (*Fan and Han, 2008*). Our ERP results indicate that BOP may provide a cognitive basis for early spontaneous neural responses to others' suffering reflected in pain expressions. Moreover, the results in Experiment five showed that the early spontaneous empathic neural responses in the P2 time window mediated the BOP effect on self-report of others' pain intensity, which further mediated the relationship between the P2 empathic responses and the amount of monetary donations. These results highlight both early spontaneous neural responses to others' pain and subjective feelings of others' pain as intermediate mechanisms by which BOP influences altruistic behavior.

To identify neural architectures underlying BOP effects on empathy, we recorded BOLD responses, using fMRI, to perceived painful and non-painful stimuli applied to individuals with patient or actor/actress identities in Experiment 6. We showed that the contrast of perceived painful (vs. non-painful) stimulations activated the sensory (i.e., post-central gyrus), affective (i.e., insula), and cognitive (i.e., mPFC) nodes of the empathy network, similar to the findings of previous studies (*Singer et al., 2004*; *Jackson et al., 2005*; *Saarela et al., 2007*; *Shamay-Tsoory et al., 2009*; *Han et al., 2009*; *Fan et al., 2011*; *Lamm et al., 2011*; *Zhou and Han, 2021*; *Luo et al., 2014*). Viewing non-painful stimulations applied to neutral faces with patient versus actor/actress identities revealed increased activity in the mPFC and bilateral TPJ, suggesting the possible neural representation of facial identities in the brain regions. Most importantly, the results of searchlight RSA that was sensitive to both stimuli and subjective feelings evoked by the stimuli revealed significant variations of activities in the insula, post-central gyrus, and lateral frontal cortex in correspondence with the patterns of self-reports of empathy for patients and actors/actresses' pain. In other words, the patterns of the activities in the insula, post-central gyrus, and lateral frontal cortex were able to predict distinct subjective feelings of patients' and actors/actresses' pain. Moreover, the results of our VPS analyses showed consistent evidence for decreased neural activities in the empathy-related neural network due to lack of BOP. These fMRI results together suggest that activities in the brain regions supporting affective sharing (e.g., insula; *Shamay-Tsoory et al., 2009*; *Fan et al., 2011*; *Lamm et al., 2019*), empathic sensorimotor resonance (e.g., post-central gyrus; *Avenanti et al., 2005*; *Zhou and Han, 2021*), and emotion regulation (e.g., lateral frontal cortex; *Ochsner and Gross, 2005*; *Etkin et al., 2015*) may provide intermediate mechanisms underlying variations of subjective feelings of others' pain intensity due to lack of BOP.

Numerous studies have shown evidence for modulations of empathy by social contexts. Contextual variables that influence the perception of others' pain and empathy include empathy targets' posture (*Martel et al., 2008*), identifiable pain pathology (*Twigg and Byrne, 2015*), moral valence (*Cui et al., 2016*; *Nicolardi et al., 2020*), and so on. Empathizers' prior exposure to pain (*Prkachin and Rocha, 2010*), socioeconomic status (*Varnum et al., 2015*), and cultural experiences (*Wang et al., 2015*; *Hampton and Varnum, 2018*) also influence empathy and its underlying brain activities. Perceived information about social relationships between observers and empathy targets also modulates empathic neural responses such that, relative to viewing own-race or own-team individuals' pain, viewing other-race or opponent-team individuals' pain decreased empathic neural responses in the affective (e.g., ACC and AI), cognitive (e.g., mPFC and TPJ), and sensorimotor (e.g., motor cortex) nodes of the empathy network (*Xu et al., 2009*; *Avenanti et al., 2010*; *Hein et al., 2010*; *Mathur et al., 2010*; *Sheng and Han, 2012*; *Sheng et al., 2014*; *Sheng et al., 2016*; *Han, 2018*; *Zhou and Han, 2021*). The perceived intergroup (racial) relationships between empathizers and empathy targets also influenced altruistic behavior such as medical treatment (*Drwecki et al., 2011*). These findings uncovered how social information perceived from stimuli and social experience modulate empathic neural responses to others' suffering and subsequent social behavior. The results of our current work complemented the findings of previous studies by uncovering how beliefs, as preexisting internal mental representations of something that is not immediately present to the scenes (*Fuentes, 2019*), also modulate people's empathy and following altruistic behavior. Specifically, in the current study, participants' beliefs (i.e., pain expressions of patients manifest their actual feelings whereas pain expressions performed by actors/actresses do not)

weakened the participants' empathy for others' pain and reduced their monetary donations to those who appeared suffering. BOP effects on empathy and altruistic behavior can be understood as modulations of empathy by preexisting internal information (e.g., beliefs) whereas previous findings revealed modulations of empathy by instantly perceived social information in a specific social context. These findings together help to construct neurocognitive models of empathy that take into consideration of both perceived social information and preexisting internal information and their interactions that lead to modulations of empathy and altruistic behavior during real-life social interactions.

It should be noted that our experimental manipulations changed the participants' minds about the models' identities (e.g., patient vs. actor/actress) rather than explicitly asking them to alter their BOP. BOP altered implicitly with target persons' identities due to observers' knowledge about individuals with different identities (e.g., painful stimuli applied to actors/actresses do not really hurt them and they show facial expressions to pretend a specific emotional state). Therefore, the BOP effects on empathy and altruistic behavior identified in our study might take place implicitly. This is different from the placebo effects on first-hand pain experiences that are produced by explicitly perceived verbal, conditioned, and observational cues that induce expectations of effective analgesic treatments (*Meissner et al., 2011*). Similar explicit manipulations of making individuals believe receiving oxytocin also promote social trust and preference for close social distances (*Yan et al., 2018*). Moreover, the placebo treatment relative to a control condition significantly attenuated activations in the ACC, AI, and subcortical structures (e.g., the thalamus) in response to painful electric shocks but increased the prefrontal activity during anticipation of painful stimulations possibly to inhibit activity in pain processing regions (*Wager et al., 2004*; *Wager and Atlas, 2015*). The brain regions in which empathic neural responses altered due to BOP (e.g., the lateral frontal cortex) as unraveled in the current study do not overlap with those in which activities are modulated by placebo analgesia (*Atlas and Wager, 2014*). These results suggest that there may be distinct neural underpinnings of BOP effects on empathic brain activity and placebo effects on brain responses to first-hand pain experiences.

Do beliefs also provide a cognitive basis for the widely documented ingroup bias in empathy for pain? Previous studies suggest that multiple neurocognitive mechanisms are involved in ingroup bias in empathy for pain such as lack of attention (*Sheng and Han, 2012*) and early group-based categorization of outgroup faces (*Zhou et al., 2020b*, see *Han, 2018* for review). There have been behavioral evidence that White individuals who more strongly endorsed false beliefs about biological differences between Blacks and Whites (e.g., 'Black people's skin is thicker than White people's skin') reported lower pain ratings for a Black (vs. White) target and suggested less accurate treatment recommendations (*Hoffman et al., 2016*). These behavioral findings suggest that other beliefs may also provide a basis for modulations of empathy for others' pain and relevant altruistic behavior. The underlying brain mechanisms, however, remain unknown. The paradigms developed in the current study may be considered in future research to examine neural underpinnings of the effects of false beliefs on empathy for pain.

Another question arising from the findings of the current study is whether the belief effect is specific to neural underpinnings of empathy for pain or is also evident for neural responses to other facial expressions. To address this issue, we conducted an additional EEG experiment in which we tested (1) whether beliefs of authenticity of others' happiness influence brain responses to perceived happy expressions, and (2) whether lack of beliefs of others' happiness also modulates neural responses to happy expressions in the P2 time window, similar to the BOP effect on ERPs to pain expressions (see Appendix 1 for methods). Similar to the paradigm used in Experiment 3, participants in the additional experiment had to first remember face identities (awardees or actors/actresses). Thereafter these faces with happy or neutral faces were presented with contextual information that the awardees showed happy expressions when receiving awards whereas actors/actresses imitated others' happy expressions. The participants also performed identity judgments on the faces while EEG was recorded. Behavioral results in this experiment showed that participants reported less feelings of actors' happiness compared to awardees' happiness. ERP results in this experiment showed that lack of beliefs of authenticity of others' happiness (e.g., actors simulating others' happy expressions vs. awardees smiling when receiving awards) reduced the amplitudes of a long-latency positive component (i.e., P570) over the frontal region in response to happy expressions. However, the face identities did not affect the P2 amplitudes in response to happy (vs. neutral)

expressions (see Appendix 1 for statistical details). These findings suggest that belief effects are evident for subjective feelings and brain activities in response to happy expressions. However, BOP or happiness affect neural responses to facial expressions in different time windows after face onset. Future research should examine neural mechanisms underlying belief effects on neural responses to other emotions to deepen our understanding of general belief effects on neural processes of others' emotional states.

Our behavioral and neuroimaging findings have implications for how we understand the general functional role of beliefs in social cognition and interaction. Empathy is supposed to originate from an evolved adaptation to quickly and automatically respond to others' emotional states during parental care that is necessary for offspring survival in humans and other species (*de Waal, 2008*; *Decety, 2011*). In most cases of interactions among family members (i.e., between parents and offspring or between siblings) perceived cues signaling pain in a person manifest his/her actual emotional states that urge help from other family members. Such life experiences may set up a default belief that perceived painful stimulation to others and their facial expressions reflect individuals' actual emotional states. This default belief provides a fundamental cognitive basis of reflexive and automatic empathy and empathic brain activity that further generates autonomic and somatic responses, as suggested by the perception-action model of empathy (*Preston and de Waal, 2002*). Nevertheless, when social interactions expand beyond family members to non-kin members and even strangers, perceived pain expressions or painful stimuli applied to others may not always manifest others' actual emotional states because perceived painful cues may be fake in some cases. BOP in such situations may function as cognitive gate-control to modulate neural responses to perceived pain in others. This is necessary for monitoring social interactions to determine whether to help or to coordinate with those who appear suffering. Our findings illustrate how the perception-emotion-behavior reactivity occurs under the cognitive constraint of BOP to keep empathy and altruistic decision/behavior for the right target who is really in need of help. In this sense, BOP also provides an important cognitive basis for survival and social adaption during social interactions.

Some limitations of the current work create future research opportunities. For example, a recent approach to hierarchical Bayesian models of cognition assumes that the brain represents information probabilistically and people represent a state or feature of the world not using a single computed value but a conditional probability density function (*Knill and Pouget, 2004*; *Friston, 2005*; *Clark, 2013*; *Tappin and Gadsby, 2019*). Our manipulations of BOP, however, had only two conditions (patient vs. actor/actress) and thus lack a model of effects of probability-based belief-updating on empathy and relevant altruistic behavior. Future research should examine how empathy and relevant altruistic behavior vary as a function of the degree of BOP. Other interesting research questions arising from our work include how the brain represents BOP. It has been proposed that different types of beliefs (e.g., empirical beliefs, conceptual beliefs, and relational beliefs) exist in the human mind and may have distinct neural underpinnings (*Harris et al., 2009*; *Seitz and Angel, 2020*). To address neural representations of BOP will allow researchers to further explore and construct neural models of the interaction between beliefs and empathic brain activity in the key nodes of the empathy network. Another interesting issue related to our findings is individual differences in BOP and BOP effects on empathy and altruism. Since specific degrees of beliefs differ widely across individuals (*Ais et al., 2016*), it is crucial to examine what personality/psychopathic traits or biological factors make individuals hold strong or weak BOP and exhibit large or small BOP effects on empathy and altruistic behavior. It is also important to clarify what environmental factors modify individuals' default BOP and consequently change their motivations to help those who appear suffering. To clarify these issues will advance our understanding of individual and contextual factors that shape the functional role of BOP in modulations of empathy and altruistic behavior. Finally, a general issue arising from the current work is whether beliefs affect the processing of other emotions such as fear, sad, and happy, and, if yes, whether there are common underlying psychological and neural mechanisms.

## Conclusion

Our behavioral and neuroimaging findings provide a new cognitive framework for understanding human empathy and altruism. Our findings indicate that lack of BOP or decreasing BOP weakened human empathy and altruistic behavior. Changing BOP affected both subjective feelings of others' emotional states and the underlying brain activity. BOP effects on altruistic behavior were

mediated by two serial mediators, that is, empathic neural responses and subjective feelings of others' pain. Our behavioral and brain imaging findings suggest that BOP provides a cognitive basis of the perception-emotion-behavior reactivity that underlies human altruism. The methods developed in our study open a new avenue for testing functional roles of beliefs as cognitive-gate control of other emotion processing and relevant social behavior.

## Materials and methods

### Participants

Sixty Chinese students were recruited in Experiment 1 as paid volunteers (29 males, mean age±s.d. =21.15±2.31 years). The sample size was estimated using G*Power (*Faul et al., 2007*) with a middle effect size of 0.25. To test the difference in pain intensity rating scores or monetary donations between the 1st_round and 2nd_round tests, we conducted ANOVAs with Test Phase (1st_round vs. 2nd_round) and Identity Change (patient to actor/actress vs. patient to patient) as independent within-subjects variables. To detect a significant Test×Identity interaction requires a sample size of 36 with an error probability of 0.05 and a power of 0.95, given the correlation among repeated measures (0.5) and the nonsphericity correction (1). Sixty Chinese students were recruited in Experiment 2 as paid volunteers (30 males, 21.55±2.45 years). Thirty Chinese students were recruited in Experiment 3 (all males, 22.23±2.51 years) as paid volunteers. The sample size was determined based on our previous EEG research on empathy for pain using the same set of stimuli (*Sheng and Han, 2012*). We recruited only male participants to exclude the potential effects of gender differences in empathic neural responses. Thirty-one Chinese students were recruited in Experiment 4 as paid volunteers. One participant was excluded from data analyses due to his lower response accuracy during EEG recording (<50%). This left 30 participants (all males, 20.70±1.97 years) for behavioral and EEG data analyses. Thirty Chinese students were recruited in Experiment 5 (all males, 20.60 ± 1.75 years). Thirty-two Chinese students were recruited in Experiment 6 as paid volunteers. One participant was excluded from data analyses due to excessive head movement during fMRI scanning. There were 31 participants left (all males, 22.23 ± 2.59 years) for behavioral and fMRI data analyses. The sample size in Experiment 6 was determined based on our previous fMRI research using similar stimuli (*Luo et al., 2014*). All participants had a normal or corrected-to-normal vision and reported no history of neurological or psychiatric diagnoses. This study was approved by the local Research Ethics Committee of the School of Psychological and Cognitive Sciences, Peking University. All participants provided written informed consent after the experimental procedure had been fully explained. Participants were reminded of their right to withdraw at any time during the study.

### Experiment 1: Lack of BOP reduces subjective estimation of empathy and altruistic behavior

#### Stimuli and procedure

The stimuli were adopted from our previous work (*Sheng and Han, 2012*), which consisted of photos of 16 Chinese models (half males) with each model contributing one photo with pain expression and one with neutral expression.

After reporting demographic information, the participants were informed that they would be paid with ¥10 as a basic payment for their participation. They would be able to obtain an extra bonus payment as much as ¥2 depending on their decisions in the following procedure. In the 1st_round test, the participants were informed that they would be shown photos with pain expressions taken from patients who suffered from a serious disease. After the presentation of each photo, the participants were asked to evaluate the intensity of each patient's pain based on his/her expression by rating on a Likert-type scale ('How painful do you think this person is feeling?', 0=not painful at all, 10=extremely painful). This rating task was adopted from previous research (*Bieri et al., 1990*; *Jackson et al., 2005*; *Lamm et al., 2007*; *Fan and Han, 2008*; *Sheng and Han, 2012*) to assess the participants' understanding of others' pain feeling—a key component of empathy. The instructions of the rating tasks focused on the emotional states of faces and had nothing to do with face identities (i.e., patients or actors/actresses). Therefore, BOP effects on empathy, if observed, occurred implicitly and automatically. Immediately after the pain intensity rating, the participants were asked

to decide how much from the extra bonus payment they would like to donate to the patient (0–10 points, one point=¥0.2). The participants were informed that the amount of one of their donation decisions would be selected randomly and endowed to a charity organization to help those who suffered from the same disease.

After the 1st_round test, the participants were asked to perform a short (5 min) calculation task (10 arithmetic calculations, e.g., 25–3×7=?) to clean their memory of the 1st_round ratings. Thereafter, the participants were told that the photos were actually taken from eight patients and eight actors/actresses and this experiment actually tested their ability to recognize social identities by examination of facial expressions. Faces assigned with patient or actor/actress identities were counterbalanced across the participants. The participants were then asked to conduct the 2nd_round test in which each photo was presented again with a word below to indicate patient or actor/actress identity of the face in the photo. The participants had to report again the pain intensity of each face and how much they would like to donate to the person shown in the photo. The participants were informed that an amount of money would be finally selected randomly from their 2nd_round decisions and donated to one of the patients through the same charity organization. After the experiments had been finished, the total amount of the participants' donations were subject to a charity organization.

We conducted ANOVAs of rating scores of pain intensity and amounts of monetary donations with Test Phase (1st_round vs. 2nd_round)×Identity Change (patient to actor/actress vs. patient to patient) as independent within-subjects variables to assess whether and how BOP influenced empathy and altruistic behavior toward those who suffered. Finally, the participants completed two questionnaires to estimate individual differences in trait empathy (*Davis, 1983*) and interpersonal trust (*Wright and Tedeschi, 1975*). We analyzed the relationship between our empathy/altruistic measures and individuals' trait empathy/interpersonal trust but failed to find significant results and thus were not reported in the main text.

## Mediation analysis

We performed mediation analyses to examine whether pain intensity mediates the pathway from BOP to monetary donation. To do this, we first dummy coded patient-identity change (i.e., 0 [patient identity in the 1st_round test] and 1 [actor/actress in the 2nd_round test]) or patient-identity repetition (i.e., as 0 [patient identity in the 1st_round test] and 1 [patient identity in the 2nd_round test]). Then, we estimated four regression models: (1) whether the independent variable (BOP) significantly accounts for the dependent variable (monetary donation) when not considering the mediator (e.g., Path c); (2) whether the independent variable (BOP) significantly accounts for the variance of the presumed mediator (pain intensity) (e.g., Path a); (3) whether the presumed mediator (pain intensity) significantly accounts for the variance of the dependent variable (monetary donation) when controlling the independent variable (BOP) (e.g., Path b); and (4) whether the independent variable (BOP) significantly accounts for the variance of the dependent variable (monetary donation) when controlling the presumed mediator (pain intensity) (e.g., Path c′). To establish the mediation, Path c is not required to be significant. The only requirement is that the indirect effect a×b is significant. Given a significant indirect effect, if Path c is not significant, the mediation is classified as indirect-only mediation which is the strongest full mediation (*Kenny et al., 1998*; *Zhao et al., 2010*). A bootstrapping method was used to estimate the mediation effect. Bootstrapping is a nonparametric approach to estimate effect-sizes and hypotheses of various analyses, including mediation (*Shrout and Bolger, 2002*; *Mackinnon et al., 2004*). Rather than imposing questionable distributional assumptions, a bootstrapping analysis generates an empirical approximation of the sampling distribution of a statistic by repeated random resampling from the available data, which is then used to calculate p-values and construct CIs. 5000 resamples were taken for our analyses. Moreover, this procedure supplies superior CIs that are bias-corrected and accelerated (*Preacher et al., 2007*; *Preacher and Hayes, 2008a*, *Preacher and Hayes, 2008b*). The analyses were performed using Hayes's PROCESS macro (Model 4; *Hayes, 2017*).

## Statistical comparison

Behavioral data were assumed to have a normal distribution but this was not formally tested. 95% CIs were reported for t-tests of the mean difference between two conditions and for correlation

analyses of correlation coefficients. 90% CIs were reported for effect sizes ($\eta_p^2$) of ANOVA analyses. According to *Steiger, 2004*, the general rule of thumb to use CIs to test a statistical hypothesis (H0) is to use a $100\times(1-\alpha)\%/100\times(1-2\alpha)\%$ CI when testing a two-sided/one-sided hypothesis at alpha level. We thus reported 90% CIs of $\eta^2$ in ANOVAs because $\eta^2$ is always positive.

## Experiment 2: Intrinsic BOP predicts subjective estimation of empathy and altruistic behavior

The face stimuli and the procedure were the same as those in Experiment 1 except the following. The participants were informed that they were to be shown photos with pain expressions taken from patients who had suffered from a serious disease and received medical treatment. After the presentation of each photo, the participants were asked to estimate how effective the medical treatment was for each patient by rating on a Likert-type scale (0=no effective or 0% effective, 100=fully effective or 100% effective). Besides rating pain intensity of each face in the 1st_round test, the participants were asked to report how unpleasant they were feeling when they viewed the photo (i.e., own unpleasantness) by rating on a Likert-type scale ('How unpleasant do you feel when viewing this person?' 0=not unpleasant at all, 10=extremely unpleasant). The unpleasantness rating was performed to evaluate emotional sharing of others' pain—another key component of empathy (*Jackson et al., 2005*; *Fan and Han, 2008*; *Sheng and Han, 2012*). The order of the two empathy rating tasks was counterbalanced across the participants. Immediately after the empathy rating tasks, the participants were asked to decide how much from the extra bonus payment they would like to donate to the patient (0–10 points, one point=¥0.2).

In the 2nd_round test, the participants were told that the medical treatment was actually effective for only half of the patients. Each photo was then presented again with information that the medical treatment applied to the patient was 100% effective or 0% effective. Thereafter, the participants were asked to perform the rating tasks and monetary donations as those in the 1st_round test. The participants were told that an amount of money would be finally selected from their 2nd_round decisions and donated to one of the patients.

### Mediation analysis

This was the same as that in Experiment 1 except that we tested whether changes of pain intensity mediate the pathway from decreased BOP or enhanced BOP to changes of monetary donation. To do this, we first calculated belief update (decreased BOP: 100% effect minus the participants' initial estimation; enhanced BOP: the participants' initial estimation minus 0% effect). Then, we estimated four regression models: (1) whether the independent variable (BOP) significantly accounts for the dependent variable (changes of monetary donation) when not considering the mediator (e.g., Path c); (2) whether the independent variable (BOP) significantly accounts for the variance of the presumed mediator (changes of pain intensity) (e.g., Path a); (3) whether the presumed mediator (changes of pain intensity) significantly accounts for the variance of the dependent variable (changes of monetary donation) when controlling the independent variable (BOP) (e.g., Path b); and (4) whether the independent variable (BOP) significantly accounts for the variance of the dependent variable (changes of monetary donation) when controlling the presumed mediator (changes of pain intensity) (e.g., Path c').

## Experiment 3: Lack of BOP decreased empathic brain activity

### Stimuli and procedure

Face stimuli were adopted from our previous work (*Sheng and Han, 2012*) and used in Experiments 3–5 in this study. The stimuli consisted of 32 faces of 16 Chinese models (half males) with each model contributed one photo with pain expression and one with neutral expression. During behavioral tests or EEG recording, each photo was presented in the center of a gray background on a 21-in. color monitor, subtending a visual angle of $3.8°\times4.7°$ (width×height: $7.94\times9.92$ cm$^2$) at a viewing distance of 60 cm.

Before EEG recording, the participants were asked to perform an identity memory task in which faces with neutral expressions were presented. Eight faces were marked as patients and eight faces as actors/actresses. After viewing photos with marked identity for 15 min, the participants performed a discrimination task in which each neutral face was displayed for 200 ms and the

participants had to press the left or right button using the left or right index finger to indicate the identity of each face (i.e., patient or actor/actress) within 2 s. After their response accuracies reached 100%, the participants were moved into an acoustically and electrically shielded booth for EEG recording.

During EEG recording, each trial consisted of a painful or neutral face with a duration of 200 ms, which was followed by a fixation cross with a duration varying randomly between 800 and 1400 ms. There were eight blocks of 64 trials (each of the 32 photographs was presented twice in a random order in each block). The participants were asked to press the left or right button using the left or right index finger to indicate the identity of the face (i.e., patient or actor/actress) as fast and accurately as possible. The relation between responding hand and face identity was counterbalanced across different blocks of trials.

After EEG recording, the participants were presented with each face again with a neutral or pain expression and asked to rate how painful the person is feeling (i.e., pain intensity) by rating on a Likert-type scale (1=not painful at all, 7=extremely painful). To estimate the participants' BOP, they were also asked to answer the question of 'To what extent do you believe the identity of this model (either patient or actor/actress)?' on a 15-point Likert-type scale (−7=extremely believed as an actor/actress, 0=not sure, 7=extremely believed as a patient).

## EEG data acquisition and analysis

A NeuroScan system (CURRY 7, Compumedics Neuroscan) was used for EEG recording and analysis. EEG was continuously recorded from 32 scalp electrodes and was re-referenced to the average of the left and right mastoid electrodes offline. Impedances of individual electrodes were kept below 5 kΩ. Eye blinks and vertical eye movements were monitored using electrodes located above and below the left eye. The horizontal electrooculogram was recorded from electrodes placed 1.5 cm lateral to the left and right external canthi. The EEG signal was digitized at a sampling rate of 1000 Hz and subjected to an online band-pass filter of 0.01–400 Hz. EEG data were filtered with a low-pass filter at 30 Hz offline. Artifacts related to eye movement or eye blinks were removed using the covariance analysis tool implemented in CURRY 7 (*Semlitsch et al., 1986*). Only trials with correct responses to face identity were included for data analyses (see *Supplementary file 15* for the numbers of trials included for data analyses in Experiments 3–5). ERPs in each condition were averaged separately offline with an epoch beginning 200 ms before stimulus onset and continuing for 1200 ms. The baseline for all ERP measurements was the mean voltage of a 200 ms prestimulus interval and the latency was measured relative to the stimulus onset.

Face stimuli in the identity judgment task elicited an early negative activity at 95–115 ms (N1) and a positive activity at 175–195 ms (P2), followed by a positive activity at 280–340 ms (P310) and a long-latency positivity at 500–700 ms (P570) over the frontal area. The mean ERP amplitudes were subject to ANOVAs with Identity (patient vs. actor/actress) and Expression (pain vs. neutral) as within-subject variables. To avoid potential significant but bogus effects on ERP amplitudes due to multiple comparisons (*Luck and Gaspelin, 2017*), the mean values of the amplitudes of the N1, P2, P310, and P570 components were calculated at frontocentral electrodes (i.e., F3, Fz, F4, FC3, FCz, and FC4).

To further assess the null hypothesis regarding the difference in the P2 amplitude in response to pain and neutral expressions of actors/actresses' faces, we conducted Bayes factor analyses for repeated-measures ANOVA and paired t-tests. We calculated the Bayes factor in the program R v.3.5.1 (http://www.r-project.org) using the function anovaBF and ttestBF from the package BayesFactor (*Morey and Rouder, 2015*). We conducted Bayes factor analyses based on the default priors for ANOVA and paired t-test design (scale r on an effect size of 0.707). A Bayes factor indicates how much more likely each alternative model is supported compared with the null hypothesis.

## Experiment 4: BOP is necessary for modulations of empathic brain activity

### Stimuli and procedure

These were the same as those in Experiment 3 except the following. Before EEG recording, the participants were informed that all the 16 faces were patients and they were from two baseball teams (half from Tiger team and half from Lion team). After the identity memory task, they performed

identity judgments on faces with neutral or pain expressions by pressing one of two buttons while EEG was recorded.

### EEG data acquisition and analysis
These were the same as those in Experiment 3.

## Experiment 5: Empathic brain activity mediates relationships between BOP and empathy/altruistic behavior

### Stimuli and procedure
The stimuli and behavioral tests were the same as those in Experiment 1 to assess BOP effects on self-report of perceived pain intensity and altruistic decisions. Thereafter, the participants went through the EEG session that was the same as that in Experiment 3 to examine BOP effects on empathic brain activity. These designs allowed us to test whether BOP-induced changes of empathic brain activity play a mediator role in the pathway from belief changes to altered subjective feelings of others' pain and altruistic decisions.

### Behavioral and EEG data recording and analyses
These were the same as those in Experiments 1 and 3.

### Multiple mediation model analysis
We constructed a serial mediation model to test the hypothesis that BOP (dummy coded as 0 for patients and 1 for actors/actresses) effect on monetary donations was sequentially mediated by two chain mediators, that is, empathic neural responses and subjective feelings of others' pain. This model includes three indirect paths: (1) indirect effect of BOP on monetary donation via empathic neural responses (i.e., P2 amplitude); (2) indirect effect of BOP on monetary donation via subjective feelings of others' pain (pain intensity); and (3) indirect effect of BOP on monetary donation via P2 amplitude×pain intensity. To do this, we estimated seven regression models: (1) whether the independent variable (BOP) significantly accounts for the dependent variable (monetary donation) when not considering the mediator (e.g., Path c); (2) whether the independent variable (BOP) significantly accounts for the variance of the presumed mediator (P2 amplitude) (e.g., Path $a_1$); (3) whether the independent variable (BOP) significantly accounts for the variance of the presumed mediator (pain intensity) (e.g., Path $a_2$); (4) whether the first independent mediator (P2 amplitude) significantly accounts for the variance of the second mediator (pain intensity) (e.g., Path $d_{21}$); (5) whether the presumed mediator (P2 amplitude) significantly accounts for the variance of the dependent variable (monetary donation) when controlling the independent variable (BOP) (e.g., Path $b_1$); (6) whether the presumed mediator (pain intensity) significantly accounts for the variance of the dependent variable (monetary donation) when controlling the independent variable (BOP) (e.g., Path $b_2$); and (7) whether the independent variable (BOP) significantly accounts for the variance of the dependent variable (monetary donation) when controlling the presumed the two mediators (e.g., Path c'). To test the significance of the three paths, we separately conducted to examine the significance of indirect effect ($a_1{\times}b_1$) of BOP on monetary donation via the P2 amplitude; indirect effect ($a_2{\times}b_2$) of BOP on monetary donation via pain intensity; and indirect effect ($a_1{\times}d_{21}{\times}b_2$) of BOP on monetary donation via P2 amplitude×pain intensity. Similarly, the bootstrapping method was used to estimate the effect size and test the hypothesis.

### Implicit association test
To assure our experimental manipulation of patient and actor/actress identities, after the EEG recording, participants were asked to complete a modified IAT (*Greenwald et al., 1998*). The participants were asked to respond to faces with patient identifies and patient-related words (e.g., ache and weak) with one key and to faces with actor/actress identities and actor/actress-related words (e.g., imitation) with another key in two blocks of trials (60 trials in each block). They were then asked to respond to faces with patient identities and actor/actress-related words with one key and to faces with actor/actress identities and patient-related words with another key in two additional blocks of trials. A D score was then calculated based on response times according to the established algorithm (*Greenwald et al., 2003*). A positive D score significantly larger than 0 would suggest that

patient faces were more strongly associated with patient (vs. actor/actress) relevant words whereas actor/actress faces were more strongly associated with actor/actress (vs. patient) relevant words.

## Experiment 6: Neural structures underlying BOP effects on empathy
### Stimuli and procedure

We adopted 24 video clips from 6 models from our previous work (*Luo et al., 2014*) and recorded 56 video clips from 14 Chinese models (half males) in Experiment 6. Each model contributed four video clips, in which a face with pain expressions receiving painful stimulation (needle penetration) or with neutral expressions receiving non-painful stimulation (cotton swab touch) applied to the left or right cheeks. Each video subtended a visual angle of 21°×17° (width×height) at a viewing distance of 80 cm during fMRI scanning.

A photo of each model with a neutral expression was obtained from each video clip. These photos were then used in the identity memory task, which was the same as that in Experiment 3. After the identity memory task, the participants underwent fMRI scanning. An event-related design was employed in six functional scans. Each scan consisted of 20 video clips (half patients [5 pain and 5 neutral expressions] and half actors/actresses [5 pain and 5 neutral expressions]) that were presented in a random order. Each video clip lasted for 3 s. There was a 9-s interstimulus interval between two successive video clips when the participants fixated at a central cross and had to judge the identity (patient or actor/actress) of each model in the video clip by pressing one of two buttons using the right index or middle finger. The relation between responding finger and face identity was counterbalanced across participants.

After fMRI scanning, the participants were presented with each video clip again outside the scanner. They were asked to rate pain intensity of each model (1=not painful at all, 7=extremely painful) and own unpleasantness (1=not unpleasant at all, 7=extremely unpleasant). Finally, we assessed the participants' beliefs of models' identities by asking them to answer the question of 'To what extent do you believe the identity of this model (either patient or actor/actress)?' on a 15-point Likert-type scale (−7=extremely believed to be an actor/actress, 0=not sure, 7=extremely believed to be a patient).

## fMRI data acquisition and analysis

Imaging data were acquired using a 3.0 T Siemens scanner with a standard head coil. Head motion was controlled to the maximum extent by using foam padding. Functional images were acquired by using T2-weighted, gradient-echo, echo-planar imaging sequences sensitive to Siemens scanner contrast (64×64×32 matrix with a spatial resolution of 3.75×3.75×5 mm$^3$, repetition time [TR]=2000 ms, echo time [TE]=30 ms, flip angle [FA]=90°, field of view=24×24 cm$^2$). Anatomical images were subsequently obtained using a standard 3D T1-weighted sequence (256×256×144 matrix with a spatial resolution of 1×1×1.33 mm$^3$, TR=2530 ms, TE=3.37 ms, inversion time=1100 ms, FA=7°).

Functional images were preprocessed using SPM12 software (the Wellcome Trust Centre for Neuroimaging, London, UK, http://www.fil.ion.ucl.ac.uk/spm). Functional scans were first corrected for within-scan acquisition time differences between slices and then realigned to the first volume to correct for inter-scan head motions. This realigning step provided a record of head motions within each fMRI run. Head movements were corrected within each run and six movement parameters (translation: x, y, z; and rotation: pitch, roll, yaw) were extracted for further analysis in the statistical model. The functional images were resampled to 3×3×3 mm$^3$ voxels, normalized to the MNI space using the parameters of anatomical normalization, and then spatially smoothed using an isotropic of 8 mm full-width half-maximum (FWHM) Gaussian kernel.

Whole-brain analyses were conducted to examine brain regions in which activities increased in response to pain versus neutral stimuli regardless of patient or actor/actress identities. This contrast pooled video clips of patient and actor/actress models together to focus on BOLD responses to painful versus neutral stimuli. The general linear model (GLM) had four regressors including patients receiving pain stimuli, patients receiving neutral stimuli, actors/actresses receiving pain stimuli, and actors/actresses receiving neutral stimuli. The GLM also included the realignment parameters to account for any residual movement-related effect. A box-car function was used to convolve with the canonical hemodynamic response in each condition. Random-effect analyses were conducted based on statistical parameter maps from each participant to allow population inference. The contrast

values were compared using whole-brain paired t-tests to identify activations, which were defined using a threshold of voxel-level p<0.001, uncorrected, cluster-level p<0.05, FWE-corrected. We also conducted a whole-brain analysis to calculate the contrast of patient versus actor/actress non-painful stimuli to test whether BOP may motivate inference of patients' mental states independently of any perceived painful cues.

## Representational similarity analysis

We conducted an RSA of brain activity (*Nili et al., 2014*) to examine neural correlates to BOP effects on subjective feelings of others' pain. We constructed a 4×4 DM for each participant with each cell in the DM represents the mean difference in rating scores of pain intensity between each pair of conditions. The DM was then used for a whole-brain searchlight RSA to identify brain regions in which the pairwise similarity of neural responses in the four conditions (two Expressions×two Identities) corresponded to the behavioral DM of condition dissimilarity in each participant. To do this, functional images were similarly preprocessed using a GLM but were not smoothed and normalized. We then estimated a GLM for each participant with Identity (patient vs. actor/actress) and Expression (pain vs. neutral) as experimental regressors. The estimated beta images corresponding to each condition were then averaged across runs at each voxel and were used as activity patterns in the RSA toolbox (*Nili et al., 2014*). We compared the neural-pattern similarity (i.e., the neural DM) with the behavioral DM in each voxel of the brain using the searchlight procedure (*Kriegeskorte et al., 2006*). The neural DM was constructed by one minus the correlation coefficient between the pattern vectors of each condition pair. The Spearman rank correlations between the neural DM and behavioral DMs were computed and assigned to the central voxel of the sphere. As such, the searchlight procedure produced Spearman p-values on each voxel for each participant, which were then subject to Fisher's z transformation for statistical tests. The resulting z maps were then normalized to standard space (resampled to 3×3×3 mm$^3$ voxels), smoothed (FWHM=8 mm), and entered into a random effect analysis using one-sample t-tests against 0. The searchlight results of all participants were then subject to a second group-level analysis to examine the voxels in the empathy network, defined based on the results of the whole-brain contrast of painful versus non-painful stimuli applied to targets, that passed a threshold of voxel-level p<0.05, FWE-corrected.

## Neural signature analysis

We conducted VPS analyses (*Krishnan et al., 2016*) to further assess BOP effects on empathic brain activity. We first calculated contrast images in the condition of patient-pain (or actor/actress-pain) versus an implicit baseline (e.g., using a design matrix of [1, 0, 0, 0]) since the test-retest reliability was higher when examining brain activations to painful stimulation using an implicit baseline than using a control condition (*Han et al., 2021*). The VPS map, which was sensitive to perceived painful stimulations applied to others' body limbs but not to self-experienced pain (*Krishnan et al., 2016*), was then converted into the image space using the ImCalc function of SPM. Thereafter, the VPS map was dot-multiplied with the contrast of patient-pain versus baseline and the contrast of actor/actress-pain versus baseline, respectively. These yielded a scalar VPS response value in each condition. The VPS response values were then subject to a one-tailed t-test to test the hypothesis of decreased VPS responses related to actor/actress-pain relative to patient-pain. To further validate the results of VPS analyses, we conducted a similar analysis using the general VPS, which was identified to respond to both perceived noxious stimulation of body limbs and painful facial expressions (*Zhou et al., 2020a*).

## Code availability

Code files used to analyze the data and to generate the figures that support the findings of this study have been uploaded.

## Acknowledgements

This work was supported by the Ministry of Science and Technology of China (2019YFA0707103) and the National Natural Science Foundation of China (projects 31871134, 31421003, and 31661143039). The authors thank the National Center for Protein Sciences at Peking University for

assistance with both experiments. The authors thank Y Zhou, T Gao, X Han, T Huo, S Mei, C Pang, Y Pu, X Wang, G Zheng, and N Zhou for proofreading the manuscript. The funder had no role in the conceptualization, design, data collection, analysis, decision to publish, or preparation of the manuscript.

## Additional information

### Funding

| Funder | Grant reference number | Author |
|---|---|---|
| Chinese Ministry of Science and Technology | 2019YFA0707103 | Shihui Han |
| Natural Science Foundation of China | 31871134 | Shihui Han |

The funders had no role in study design, data collection and interpretation, or the decision to submit the work for publication.

### Author contributions

Taoyu Wu, Data curation, Software, Formal analysis, Validation, Visualization, Methodology, Writing - original draft, Writing - review and editing; Shihui Han, Conceptualization, Formal analysis, Supervision, Funding acquisition, Investigation, Visualization, Methodology, Writing - original draft, Project administration, Writing - review and editing

### Author ORCIDs

Shihui Han (iD) https://orcid.org/0000-0003-3350-5104

### Ethics

Human subjects: This study was approved by the local Research Ethics Committee of the School of Psychological and Cognitive Sciences, Peking University. All participants provided written informed consent after the experimental procedure had been fully explained. Participants were reminded of their right to withdraw at any time during the study.

### Decision letter and Author response

Decision letter https://doi.org/10.7554/eLife.66043.sa1
Author response https://doi.org/10.7554/eLife.66043.sa2

## Additional files

### Supplementary files

- Source code 1. Scripts for plotting *Figures 1a, b*, *2d, e, f*, *3a*, *4a*, *5a and b*.

- Source code 2. Scripts for plotting *Figures 3c*, *4c* and *5d*.

- Source code 3. Scripts for the whole-brain analysis in *Figure 6a and b*.

- Source code 4. Scripts for plotting *Figure 6c*.

- Source code 5. Scripts for plotting *Figure 6d*.

- Supplementary file 1. Statistical results of the mediation analysis (pain intensity mediated the relationship between decreased BOP and monetary donations) in Experiment 1.

- Supplementary file 2. Statistical results of the mediation analysis (pain intensity mediated the relationship between enhanced BOP and monetary donations) in Experiment 1.

- Supplementary file 3. Pain intensity, unpleasantness, and monetary donation (mean±SD) in Experiment 2.

- Supplementary file 4. Statistical results of the mediation analysis (pain intensity mediated the relationship between decreased BOP and monetary donations) in Experiment 2.

- Supplementary file 5. Statistical results of the mediation analysis (pain intensity mediated the relationship between enhanced BOP and monetary donations) in Experiment 2.
- Supplementary file 6. Statistical results of the mediation analysis (unpleasantness mediated the relationship between decreased BOP and monetary donations) in Experiment 2.
- Supplementary file 7. Statistical results of the mediation analysis (unpleasantness mediated the relationship between enhanced BOP and monetary donations) in Experiment 2.
- Supplementary file 8. Statistical results of reaction times, accuracies, and rating scores (mean±SD) in Experiment 3.
- Supplementary file 9. Statistical results of mean ERP amplitudes (mean±SD) in Experiment 3.
- Supplementary file 10. Statistical results of reaction times, accuracies, and rating scores (mean±SD) in Experiment 4.
- Supplementary file 11. Statistical results of mean ERP amplitudes (mean±SD) in Experiment 4.
- Supplementary file 12. Statistical results of reaction times, accuracies, and mean ERP amplitudes (mean±SD) in Experiment 5.
- Supplementary file 13. Results of the serial mediation analysis in Experiment 5.
- Supplementary file 14. Statistical results of reaction times, accuracies and rating scores (mean±SD) in Experiment 6.
- Supplementary file 15. Number of ERP trials for analyses (mean±SD) in Experiments 3–5.
- Transparent reporting form

### Data availability
Source data files have been provided for Figures 1-6 and Appdendix 1 Figure 1.

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

## Appendix 1

Our results in Experiments 1–6 showed consistent evidence for modulations of both subjective (self-report) and objective (EEG/fMRI) measures of empathy for others' suffering. An interesting question arising from these findings is whether the belief effects are specific to neural underpinnings of empathy for pain. We addressed this issue by examining belief effects on neural responses to other facial expressions in an additional experiment. Specifically, in this experiment, we sought to test (1) whether beliefs of authenticity of others' happiness influence brain responses to perceived happy expressions and (2) whether beliefs also modulate neural responses to happy expressions in the P2 time window, similar to the BOP effect on ERPs to pain expressions. The paradigm used in the additional experiment was the same as that used in Experiment 3 except the following. We asked an independent sample of participants to remember identities (awardees or actors/actresses) of neutral faces. Thereafter, EEG signals to happy and neutral expressions of awardees or actors/actresses were recorded after informing the participants that photos of happy faces were taken from awardees who were smiling when receiving awards whereas actors/actresses imitated others' smiling and showed happy expressions. We predicted that beliefs that actors/actresses' expressions do not reflect their actual emotional states would decrease brain response to happy expressions. We tested this prediction by comparing ERPs to happy/neutral faces with awardee or actor/actress identities.

We recorded EEG signals from an independent sample of healthy young adults (N=30 males, mean age±s.d.=22.30±2.73 years). Face stimuli with happy or neutral expressions were adopted from the previous study (*Wang and Han, 2021*). There were photos of 16 Chinese models (half males) and each model contributed one photo with happy expression and one with neutral expression.

The participants were first presented with the faces with neutral expressions and were informed that these photos were taken from eight awardees who recently obtained awards and from eight actors/actresses. After the identity memory task, in which the participants were able to correctly recognize all faces with awardee or actor/actress identities, they were asked to perform identity judgments on faces with neutral or happy expressions by pressing one of two buttons while EEG was recorded. After EEG recording, the participants were presented with each happy face again and had to rate how happy the person is feeling (i.e., happiness intensity) by rating on a Likert-type scale (1=not happy at all, 7=extremely happy).

An ANOVA of the mean rating scores of happiness intensity with Identity (awardee vs. actor/actress) and Expression (happy vs. neutral) as within-subject variables revealed significant main effects of Identity (F(1,29)=19.512, p<0.001, $\eta_p^2$=0.402, 90% CI=[0.166, 0.560]) and Expression (F(1,29) =422.774, p<0.001, $\eta_p^2$=0.936, 90% CI=[0.889, 0.953]), and a significant Identity×Expression interaction (F(1,29)=6.610, p=0.016, $\eta_p^2$=0.186, 90% CI=[0.021, 0.372], see *Appendix 1—figure 1a*, and *Appendix 1—table 1* for details). The results suggest weaker subjective feelings of happiness intensity for faces with actor/actress identities compared to awardee identities.

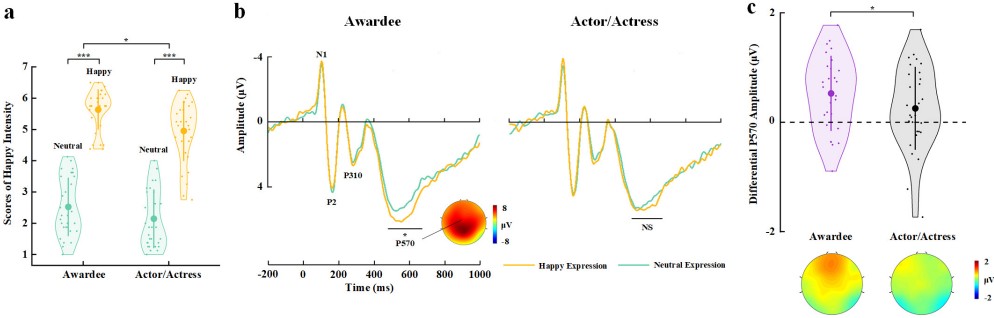

**Appendix 1—figure 1.** EEG results of the additional experiment. (**a**) Mean rating scores of happy intensity related to happy and neutral expressions of faces with awardee or actor/actress identities. (**b**) ERPs to faces with awardee or actor/actress identities at frontal electrodes. The voltage topography shows the scalp distribution of the P570 amplitude with the maximum over the central/

*Appendix 1—figure 1 continued on next page*

*Appendix 1—figure 1 continued*

parietal region. (**c**) Mean differential P570 amplitudes to happy versus neutral expressions of faces with awardee or actor/actress identities. The voltage topographies illustrate the scalp distribution of the P570 difference waves to happy (vs. neutral) expressions of faces with awardee or actor/actress identities, respectively. Shown are group means (large dots), standard deviation (bars), measures of each individual participant (small dots), and distribution (violin shape) in (**a**) and (**c**). EEG, electroencephalography; ERP, event-related potential.

The online version of this article includes the following source data is available for figure 1:

**Appendix 1—figure 1—source data 1.** Happy intensity rating scores and Mean differential P570 amplitudes.

**Appendix 1—table 1.** RTs, accuracies, rating scores, numbers of ERP trials, and ERP amplitudes (mean±SD) in the additional experiment.

| | | Awardee | | Actor/Actress | |
|---|---|---|---|---|---|
| | | **Neutral** | **Happy** | **Neutral** | **Happy** |
| RT (ms) | | 654±63 | 657±60 | 666±64 | 680±66 |
| Accuracy (%) | | 92±4.9 | 90±7.5 | 92±5.4 | 88±8.7 |
| Happy intensity | | 2.525±0.94 | 5.638±0.64 | 2.146±0.94 | 4.95±0.96 |
| N1 amplitude (µV) | | −2.267±1.69 | −2.606±1.75 | −2.297±1.43 | −2.620±1.52 |
| P2 amplitude (µV) | | 2.544±2.64 | 2.375±2.30 | 2.940±2.56 | 2.593±2.56 |
| P310 amplitude (µV) | | 3.449±3.45 | 3.445±3.30 | 3.492±3.38 | 3.376±3.38 |
| P570 amplitude (µV) | | 4.677±2.22 | 5.379±2.15 | 4.696±2.16 | 4.950±2.11 |
| ERP trials | | 114±10 | 110±13 | 113±11 | 108±12 |
| | Statistic value | ANOVA | | | Simple effect (Identity) | |
| | Value | Identity | Expression | Identity*Expression | Awardee | Actor/Actress |
| RT (ms) | F | 13.229 | 11.256 | 4.733 0.7 | 0.915 | 13.230 |
| | p | 0.001 | 0.002 | 0.038 | 0.347 | 0.001 |
| | $\eta^2_P$ | 0.313 | 0.280 | 0.140 | 0.031 | 0.313 |
| | 90% CI | (0.094, 0.488) | (0.071, 0.459) | (0.004, 0.326) | (0, 0.180) | (0.094, 0.488) |
| Accuracy (%) | F | 0.496 | 40.590 | 0.595 | | |
| | p | 0.487 | <0.001 | 0.447 | | |
| | $\eta^2_P$ | 0.017 | 0.583 | 0.020 | | |
| | 90% CI | (0, 0.150) | (0.362, 0.698) | (0, 0.158) | | |
| Happy Intensity | F | 19.512 | 422.774 | 6.610 | 433.364 | 302.128 |
| | p | <0.001 | <0.001 | 0.016 | <0.001 | <0.001 |
| | $\eta^2_P$ | 0.402 | 0.936 | 0.186 | 0.937 | 0.912 |
| | 90% CI | (0.166, 0.560) | (0.889, 0.953) | (0.021, 0.372) | (0.892, 0.955) | (0.849, 0.937) |
| N1 (95–115 ms) | F | 0.031 | 9.890 | 0.005 | | |
| | p | 0.862 | 0.004 | 0.944 | | |
| | $\eta^2_P$ | 0.001 | 0.254 | 0.0002 | | |
| | 90% CI | (0, 0.041) | (0.055, 0.436) | (0, 0.007) | | |

*Continued on next page*

*Appendix 1—table 1 continued*

| | | Awardee | | | Actor/Actress | |
| --- | --- | --- | --- | --- | --- | --- |
| | | Neutral | Happy | | Neutral | Happy |
| P2 (175–195 ms) | F | 6.476 | 2.822 | 0.441 | | |
| | p | 0.017 | 0.104 | 0.512 | | |
| | $\eta_p^2$ | 0.183 | 0.089 | 0.015 | | |
| | 90% CI | (0.019, 0.369) | (0, 0.266) | (0, 0.145) | | |
| P310 (280–340 ms) | F | 0.012 | 0.140 | 0.252 | | |
| | p | 0.913 | 0.711 | 0.619 | | |
| | $\eta_p^2$ | 0.0004 | 0.005 | 0.009 | | |
| | 90% CI | (0, 0.017) | (0, 0.106) | (0, 0.125) | | |
| P570 (500–700 ms) | F | 1.948 | 20.752 | 4.832 | 20.880 | 3.375 |
| | p | 0.173 | <0.001 | 0.036 | <0.001 | 0.076 |
| | $\eta_p^2$ | 0.063 | 0.417 | 0.143 | 0.419 | 0.104 |
| | 90% CI | (0, 0.232) | (0.180, 0.572) | (0.005, 0.328) | (0.181, 0.573) | (0, 0.285) |

Note: Effect size is indexed as the partial eta-squared value. The 90% CIs are reported for partial eta-squared value.

The participants responded to face identities with high accuracies during EEG recording (>88% across all conditions, see *Appendix 1—table 1* for details). Similarly, ERPs to face stimuli in this experiment were characterized by an early negative activity at 90–120 ms (N1) and a positive activity at 175–195 ms (P2) at the frontal/central regions, which were followed by two positive activities at 280–340 ms (P310) over the parietal region and 500–700 ms (P570) over the frontal area (*Appendix 1—figure 1b*). ANOVAs of the P2 amplitudes with Identity (awardee vs. actor/actress) and Expression (happy vs. neutral) as within-subject variables did not reveal a significant Identity-×Expression interaction (F(1,29)=0.441, p=0.512, $\eta_p^2$=0.015, 90% CI=[0, 0.145], Bayes factors=0.303).

Importantly, ANOVAs of the later P570 amplitudes showed a significant Identity×Expression interaction (F(1,29)=4.832, p=0.036, $\eta_p^2$=0.143, 90% CI=[0.005, 0.328], *Appendix 1—figure 1b and 1c*, see *Appendix 1—table 1* for statistical details). Simple effect analyses indicated significantly larger P570 amplitudes in response to happy versus neutral expressions of awardees' faces (F(1,29)=20.880, p<0.001, $\eta_p^2$=0.419, 90% CI=[0.181, 0.573]), but not of actors/actresses' faces (F(1,29)=3.375, p=0.076, $\eta_p^2$=0.104, 90% CI=[0, 0.285], Bayes factor=0.858).

Our behavioral and ERP results in this experiment suggest reduced subjective feelings and brain responses to happy (vs. neutral) expressions of actors/actresses' faces compared to awardees' faces. These results support the prediction that beliefs that actors/actresses' expressions do not reflect their actual emotional states decrease brain response to happy expressions. However, belief effects on brain responses to happy expressions were observed on the P570 amplitudes but not on the P2 amplitudes. This is different from our ERP results in Experiments 3–5, in which we showed evidence that BOP modulated the P2 amplitudes. These results suggest general belief modulation effects on brain activities involved in processing of facial expressions. In addition, our results suggest that the time window in which beliefs modulate brain responses to facial expressions depends on the nature of facial expressions (e.g., pain or happiness expressions).

