## [Decision Letter]

**Acceptance summary:**

This article represents a series of behavioral and imaging experiments investigating the effect of cognitive manipulation of beliefs on the explicit perception of other individual's pain and altruistic behavior. The results indicate that manipulations of people's beliefs regarding how much another person suffers alters the way participants rate the pain of others as well as the amount of money participants are willing to donate to the other person. Neuroimaging experiments, using EEG and fMRI, show that manipulating beliefs modulates neural activity in an early time window (P2 component) in response to the emotional expression of the other individual, involving temporo-parietal, post-central, insular, and frontal cortices. The results are overall clear and consistent throughout the six experiments, integrate with existing data and are of broad interest to the researchers studying the neurobiology of empathy.

**Decision letter after peer review:**

Thank you for submitting your article "Neural mechanisms underlying regulation of empathy and altruism by beliefs of others' pain" for consideration by *eLife*. Your article has been reviewed by 2 peer reviewers, and the evaluation has been overseen by a Reviewing Editor and Floris de Lange as the Senior Editor. The reviewers have opted to remain anonymous.

Essential Revisions (for the authors):

1) Framing and interpretation of the data.

Both reviewers question whether some of the data are correctly interpreted. This will require either additional evidence (new data) or a reframing of some of the reported results (particularly the pain intensity ratings).

2) Couching of the current results in the literature.

Both reviewers make several recommendations how to better connect your current study to the previous extensive literature on this topic.

3) Methodological issues.

The reviewers raise several methodological issues that will need to be clarified in a revised version of the paper.

*Reviewer #1 (Recommendations for the authors):*

I really appreciate and value the effort and amount of work the authors put in the six experiments. The method and analyses are clear and do not present major concerns. The results of the six experiment nicely integrate and replicate.

My main concern is in the interpretation of the results. Below I will try in detail to explain the problems I see.

In most of the experiments, participants are told the half of the pictures depict actors reproducing painful facial expressions. By doing so, the experimenter is basically telling participants that half of the pictures show a "fake" expression of pain, in other words, participant in the second round now know that the actor does not feel any pain. That participants' ratings of pain go down is then perhaps rather trivial. That the ratings do not go down to zero might instead be more interesting. However, asking to rate in how much pain an actor is that pretends to be in pain is an odd question to begin with. Rationally, one knows the actor has no somatic pain. So, a nuanced analysis of what the participants believe the experimenter to want them to report would be essential here, but appears to be missing. Do participants take the question to probe how much pain they now perceive to be expressed in the facial expression (disregarding their knowledge) or how much negative emotions the actors conjured in his mind to generate authentic facial expressions, or how much somatic pain the actor feels, or an average of all these? I find it rather frustrating, for the interpretation of the results, to not quite know how the participants interpret this question in the second session. By detaching the facial expressions from somatic pain through the belief manipulation, the authors create a situation in which reported painfulness becomes a question of reporting abstract *beliefs*, much as one would in typical false belief tasks, rather than reporting sensory perception. Similarly, asking about the unpleasantness experienced by the participant now becomes a compound report. Is it unpleasant to know that an emotion was faked? Is a negative feeling now reflecting empathy with pain, a feeling of being lied to, or a combination of both? (similar reasoning apply for the treatment manipulation, in which it is not clear whether participants now feel empathy or pity or something else). The authors continue to equate these measures with pain empathy, but I find it difficult to take that equation at face value.

The core of the issue is then what the paper is about. If the paper remains entirely in the domain of beliefs, and the question is whether telling people that someone doesn't really feel pain allows them to report the belief that the person is in less pain than originally thought, I think the data is sound, but the conclusion somewhat trivial. If the question is whether telling someone that an actor doesn't really experience pain reduces an automatic affective empathy for pain, which then alters reports of perceived pain, I do not see how the data would inform that question. There is no measure in the report that can be taken as a clear measure of affective empathy: it is unclear whether the rating reflects cognitive empathy or other cognitive processes (see questions above), the N2 is not only present when witnessing pain, but is typical of many cognive control tasks and the fMRI data reflecting the modulations are areas involved in cognitive tasks and beliefs beyond pain.

I will give an example of an experiment that could help explaining my thinking. Imaging an experiment in which I tell participants to rate how green is the apple I show them. After the rating I tell participants the apple was actually yellow, but was painted green before the experiment. Now I ask them again to rate the greenness. What do we expect participants to do? To rate the color they actually see (green) or to rate what they now know to be the real color of the apple (yellow), the one they cannot see? Without clear specification, some participants will interpret the question as to rate the color they know, and others to rate color they see. In both cases, the average rating might decrease, but would this decrease allow us to say that the cognitive manipulation has permeated the early stages of colour perception? Without knowing how participants interpreted the question, the best we would be able to say is that the cognitive manipulation modulated the final color report.

Additional remarks:

– It is not clear to me why the authors refer to the patient-to-patient condition as an enhancement of BOP. The results do indeed suggest that the rating and donation increases, but the manipulation was not meant to enhance the BOP, but to stay stable (an example of enhanced BOP would be if participants would have been told that in the first run they saw actors, but then discovered they were actually patients). Authors cannot define their manipulation based on the results. In all their experiments, one identity remains the same, and only the other changes, therefore BOP is expected to remain the same in one case and decrease (or increase) in the other condition. I would advise to rename their BOP accordingly, and not based on the results.

– The authors focus on the N2 components, which makes the manuscript nicely focused. The rationale of this choice is though not very clear, as the literature suggests other components also to be modulated by task instructions (see for instance the following meta-review https://academic.oup.com/scan/article/13/10/1003/5077584?login=true). Authors should justify the focus or report the results for the other components. As mentioned above, authors should also be careful in equating N2 as pain-specific component.

– The ANOVA on N2 shows a nice Identity x Expression interaction, which the authors interpret as a reduction of empathic neural responses, as the difference between pain and neutral is bigger while watching patient's facial expression. Beside the already mentioned problem of taking N2 as a marker of pain empathy, by looking at the graph I seem to notice that the change in the difference, could be more driven by a change in the N2 amplitude in the response to neutral stimuli (increase of N2 amplitude while watching the actors), rather than a reduction of N2 to the painful stimuli, which seems to stay stable across conditions. If this were indeed the case, it would again suggest that what might have been modulated is not per-se the specific response to pain, but to something like saliency (someone not saying the truth becomes salient) or trust.

– The manipulation of the team-identity in Exp4 is weak. Of course, it is good that it does not change the believe of how much the different team-mate should feel pain (which is what needed indeed for such a control experiment), but unfortunately it also doesn't create a systematic bias of any kind. As it stands, it boils down to a lable permutation of the identities, which one would not expect to play any modulatory role on any beliefs. If the goal was to test the specificity for pain of the BOP, the authors should have chosen a believe manipulation that was not affecting the pain domain, but another domain, for instance choosing a team the participants was a fan of, and a team that participants do not care about or does not like. By choosing two irrelevant teams, the observed lack of an interaction is trivial, and the authors cannot conclude that BOP is necessary for the modulation of empathic activity.

– In Exp6 it is not clear why the authors do not present the results of the interaction, which is a central contrast throughout the manuscript. Again, the authors lack evidence for a modulation of pain-empathy. What the authors could do to tap into the questions of whether their BOP modulation affects empathy for pain, is to use some form of multivariate pattern that has higher specificity for pain processing. One way the author could try to use to be able to say something about the involvement of empathy for pain is by using the vicarious pain signature maps (Somatic and vicarious pain are represented by dissociable multivariate brain patterns *eLife* (elifesciences.org)). Author could multiply the VPS maps with their β weights from the SPM model for the four conditions: patient-actor pain, patient-actor neutral, patient-patient pain, patient-patient neutral, and run an ANOVA on the resulting values. In this way they could basically test the interaction within a network that has been reliably associated with pain-empathy, and the validity of which has been tested across experiments.

Here below a list of some of the statements unsupported by the data:

– "Manipulated BOP changes empathy and altruistic behavior". As I tried to explain, there is no clear measure of empathy, due to the interpretability of the tasks from the participants' point of view. The statement is correctly supported in the case of donation. The concern is only about empathy.

– "Intrinsic BOP predicts empathy and …". Again it is not clear what participants are reporting and interpreting after they know that the treatment did not have an effect (pity, despair, helplessness).

– "Manipulated BOP changes empathic brain activity". The presence of a N2 is not sufficient to claim that there is empathic brain activity, as N2 can be elicited by salient stimuli as well (even the fact that there is an N2 in response to the neutral stimuli, suggests that N2 might represent other activity than empathic).

– "Together, the results in Experiment 3 and 4 suggest a key role of BOP, but not of other cognitive processes involved in the experimental manipulations, in modulations of neuronal responses to other's pain." The control experiment does not contain any other manipulation of cognitive beliefs, and therefore the authors cannot exclude that other beliefs not related to pain could have equally modulated the N2 component.

– "Our behavioral and neuroimaging results revealed critical functional roles of BOP in modulation of the perception-emotion-behavior reactivity by showing how BOP predicted and affected empathy/empathic brain activity and monetary donation. Our finding provide evidence that BOP constitutes a fundamental cognitive basis for empathy and altruistic behavior." The neuroimaging results revealed that areas involved in cognitive processes are modulated by cognitive manipulation. There is no evidence that empathy-related neuronal activity was modulated by the BOP manipulation. BOP is a successful cognitive manipulation, but what authors cannot conclude that BOP is a fundamental cognitive basis for empathy.

– "These findings together support that empathy, as either an instant emotional response to other's suffering or a sustained personality trait of understanding and sharing other's pain, is one of the key motivation factors for altruistic behavior." I do not see how the results support such a claim, and there is no measure of sustained personality trait of understanding either.

*Reviewer #2 (Recommendations for the authors):*

Experiment 1 follow-up notes:

– As I noted in my public review, the "enhanced" responses towards the targets appearing as Patients in both Rounds 1 and 2 could simply be a contrast effect due to the Patient-Actor targets. I think there are a number of ways the authors could rule out this possibility. For example, you could show just 8 patients in Round 1, followed by 16 patients in Round 2 (8 new, 8 returning from Round 1). Alternatively, you could show 8 actors/actresses and 8 patients in Round 1, and then tell participants that *all* these targets are actually patients in Round 2. My prediction in both cases would be that you *wouldn't* see an increase in empathy/altruism/etc. for the targets appearing as patients in both rounds in either case.

– Moreover, I referred to some confusion about the charitable donations in this study. In Round 1, the charitable donation is described like this: "The participants were informed that the amount of one of their decisions would be selected randomly and endowed to a charity organization to help those who suffered from the same disease. " In Round 2, it's described like this: "The participants had to…report how much they would like to donate from the extra bonus payment to each model. The participants were told that an amount of money would be finally selected from their 2^nd^ round donation decisions and presented to a charity organization after the study." I wasn't sure if the difference in wording (e.g., "a charity organization to help those who suffered from the same disease" vs. just "a charity organization") was important and intentional – is it the same disease-specific charity in both rounds? (And is it the same charity for both target types in Round 2 – e.g., both patients and actors?) If the answer to either of these questions is no, more detail is necessary, since this could potentially explain differences in giving across condition. (Also, if participants know that these targets are not actually receiving their money, but rather that a trial will be selected at random to determine their donation, it's a little surprising that participants would vary their responses at all – IF the money goes to the same charity regardless of target type.)

Experiment 2 typographical notes:

– In lines 274-277, the authors report the results for intensity and unpleasantness ratings and donations in a manner that's a little hard to follow ("One-sample t-tests showed that the z values were significantly smaller than zero for decreased-BOP patients (correlations between changes in belief and pain intensity/unpleasantness/monetary donation: mean {plus minus} s.d. = -0.304 {plus minus} 0.370, -0.277 {plus minus} 0.455 and -0.236 {plus minus} 0.410; t(59) = -6.352, -4.706 277 and -4.465; ps < 0.001; Cohen's d = 0.822, 0.609 and 0.576; 95% CI = (-0.400, -0.208), 278 (-0.394, -0.159), and (-0.342, -0.130))") – I'd recommend splitting up these measures, rather than grouping by statistical value (e.g., all three means, all three p-values, etc.).

Experiment 4 open-ended question:

– This study *did* get me thinking about the considerable literature on intergroup empathy that uses minimal groups or real-world group affiliations (e.g., soccer teams, etc.) to demonstrate that people show an in-group bias in empathy for pain. How should we conceptualize those findings in terms of BOP? (e.g., Are differential expectancies about pain tolerance/sensitivity/etc. embedded in these group memberships?) Or are those group-based effects supported by fundamentally different mechanisms?

---

## [Author Response]

Essential Revisions (for the authors):1) Framing and interpretation of the data.Both reviewers question whether some of the data are correctly interpreted. This will require either additional evidence (new data) or a reframing of some of the reported results (particularly the pain intensity ratings).

Thanks for these suggestions. Accordingly, we conducted a new EEG experiment (i.e., Supplementary Experiment 1 in the revision) and reported new EEG data to examine belief effects on neural responses to happy expressions to address Reviewer #1's question regarding whether belief effects are specific to neural processing of pain expressions. We also reported new fMRI results of RSA and VPS analyses in response to Reviewer #1's comments and suggestions (page 37-39). We discussed the implications of our new EEG and fMRI results in the revised Discussion (page 43-44, 45-47). We also cited previous studies of empathy for pain in the revised Introduction to explain how researchers employed subjective (self-report of rating scores of others' pain intensity and own unpleasantness) and objective (EEG/fMRI) measures of empathy (page 6-7). We believe that these modifications help to reframe our results of both subjective estimation and objective estimation of empathy and BOP effects on empathy for pain.

2) Couching of the current results in the literature.Both reviewers make several recommendations how to better connect your current study to the previous extensive literature on this topic.

In the revised Introduction we cited additional literatures of studies of pain expression to connect our work with previous findings (page 7-8). In the revised Discussion we included one paragraph to connect our work with previous findings regarding how empathy is modulated by preexisting internal information (e.g., beliefs) and by instantly perceived social information (page 46-47), one paragraph regarding the relationship between our brain imaging findings and previous behavioral findings of false belief effects on empathy and altruistic behavior (page 48-49), and one paragraph to discuss general effects of beliefs on neural processing of facial expressions (page 49-50).

3) Methodological issues.The reviewers raise several methodological issues that will need to be clarified in a revised version of the paper.

We have now modified the Method section to clarify the reviewers’ concerns about measures in the first and second round tests (page 55-56). We also included an additional paragraph in the Method section about VPS analysis of fMRI signals (page 71-72).

Reviewer #1 (Recommendations for the authors):I really appreciate and value the effort and amount of work the authors put in the six experiments. The method and analyses are clear and do not present major concerns. The results of the six experiment nicely integrate and replicate.My main concern is in the interpretation of the results. Below I will try in detail to explain the problems I see.In most of the experiments, participants are told the half of the pictures depict actors reproducing painful facial expressions. By doing so, the experimenter is basically telling participants that half of the pictures show a "fake" expression of pain, in other words, participant in the second round now know that the actor does not feel any pain. That participants' ratings of pain go down is then perhaps rather trivial. That the ratings do not go down to zero might instead be more interesting. However, asking to rate in how much pain an actor is that pretends to be in pain is an odd question to begin with. Rationally, one knows the actor has no somatic pain. So, a nuanced analysis of what the participants believe the experimenter to want them to report would be essential here, but appears to be missing. Do participants take the question to probe how much pain they now perceive to be expressed in the facial expression (disregarding their knowledge) or how much negative emotions the actors conjured in his mind to generate authentic facial expressions, or how much somatic pain the actor feels, or an average of all these? I find it rather frustrating, for the interpretation of the results, to not quite know how the participants interpret this question in the second session. By detaching the facial expressions from somatic pain through the belief manipulation, the authors create a situation in which reported painfulness becomes a question of reporting abstract beliefs, much as one would in typical false belief tasks, rather than reporting sensory perception. Similarly, asking about the unpleasantness experienced by the participant now becomes a compound report. Is it unpleasant to know that an emotion was faked? Is a negative feeling now reflecting empathy with pain, a feeling of being lied to, or a combination of both? (similar reasoning apply for the treatment manipulation, in which it is not clear whether participants now feel empathy or pity or something else). The authors continue to equate these measures with pain empathy, but I find it difficult to take that equation at face value.The core of the issue is then what the paper is about. If the paper remains entirely in the domain of beliefs, and the question is whether telling people that someone doesn't really feel pain allows them to report the belief that the person is in less pain than originally thought, I think the data is sound, but the conclusion somewhat trivial. If the question is whether telling someone that an actor doesn't really experience pain reduces an automatic affective empathy for pain, which then alters reports of perceived pain, I do not see how the data would inform that question. There is no measure in the report that can be taken as a clear measure of affective empathy: it is unclear whether the rating reflects cognitive empathy or other cognitive processes (see questions above), the N2 is not only present when witnessing pain, but is typical of many cognive control tasks and the fMRI data reflecting the modulations are areas involved in cognitive tasks and beliefs beyond pain.I will give an example of an experiment that could help explaining my thinking. Imaging an experiment in which I tell participants to rate how green is the apple I show them. After the rating I tell participants the apple was actually yellow, but was painted green before the experiment. Now I ask them again to rate the greenness. What do we expect participants to do? To rate the color they actually see (green) or to rate what they now know to be the real color of the apple (yellow), the one they cannot see? Without clear specification, some participants will interpret the question as to rate the color they know, and others to rate color they see. In both cases, the average rating might decrease, but would this decrease allow us to say that the cognitive manipulation has permeated the early stages of colour perception? Without knowing how participants interpreted the question, the best we would be able to say is that the cognitive manipulation modulated the final color report.

Many thanks for these important and detailed comments. We believe that there are two important points here. First, what is the central research question of our study? In the revised Introduction we clarified that our study tested the hypothesis that belief of others' pain (BOP) provides a cognitive basis for empathy and altruistic behavior by modulating brain activity in response to others' pain. Specifically, we tested whether lack of BOP or weakening BOP results in inhibition of altruistic behavior by decreasing empathy and its underlying brain activity (page 5-6). We explained the design of our experiments in details by clarifying that our behavioral and EEG/fMRI experiments were designed based on the common beliefs that patients show pain expressions to manifest their actual feelings of pain whereas pain expressions performed by actors/actresses do not indicate their actual emotional states (page 8-10). We clarified that different beliefs were built during a learning procedure that assigned different identities (e.g., patient or actor/actress) to faces. We measured participants' self-reports of others' pain and own unpleasantness and brain responses related to the same sets of faces with pain or neutral expressions, which, however, were informed to be patients or actors/actresses. During EEG/fMRI recording the participants were asked to discriminate patient or actor/actress identities to activate different beliefs, i.e., patients show pain expressions to manifest their actual feelings of pain whereas pain expressions performed by actors/actresses do not indicate their actual emotional states. The revised Introduction clarified that we compared self-reports and brain activities related to pain (vs. neutral) expressions of actors/actresses' faces relative to those of patients' faces to assess whether lack of BOP reduced altruistic behavior and whether BOP effects were mediated by decreased empathy due to weakened BOP.

The second point is whether self-reports of rating scores used in our study.

measured empathy and whether self-report ratings used to estimate empathy are sufficient for addressing the research question of our study. In the revised Introduction, we addressed these issues by providing details of how previous studies conducted subjective and objective estimations of empathy for others' pain (page 6-7). Specifically, we clarified that previous studies collected self-reports of others' pain and own unpleasantness when observing others' pain as a subjective estimation of empathy for pain (e.g., Bieri et al., 1990; Jackson et al., 2005; Lamm et al., 2007; Fan and Han, 2008; Sheng and Han, 2012). These two measures correspond to understanding and sharing the two components of empathy.

However, researchers realized that self report can be influenced by social contexts and depend on participants' understanding of task instructions. Therefore, it is necessary to estimate empathy by recording brain activities that differentially respond to painful vs. non-painful stimuli applied to others (or pain vs. neutral expressions) as an objective estimation of empathy for pain (e.g., Singer et al., 2004; Jackson et al., 2005; Gu and Han, 2007; Fan and Han, 2008; Hein et al., 2010; Sheng and Han, 2012). More importantly, fMRI studies revealed brain regions such as the ACC, AI, sensorimotor cortices, and temporoparietal junction, which are involved in empathy and predict rating scores related to understanding (or cognitive) or sharing (affective) of others' pain. It is now widely agreed that measuring rating scores others' pain and own unpleasantness is not sufficient to assess empathy and brain imaging provides an objective estimation of empathy. This is why, in our study, we reported four EEG/fMRI experiments to address our research questions even though we first reported two behavioral experiments. Our Experiments 1 and 2 gave participants clear task instructions to report on a scale how painful others feel and how unpleasant the participants feel when viewing others' pain, as subjective estimation of empathy for pain, similar to numerous previous studies. The instructions of the rating tasks focused on emotional states of faces and had nothing to do with face identities (i.e., patients or actors). Under these task instructions, BOP effects on empathy in both our behavioral and EEG/fMRI experiments occurred implicitly and automatically. In our revised Method we made clear this point (page 55-56). As mentioned above, we examined BOP effects on empathy by comparing self-reports and brain activities related to pain (vs. neutral) expressions of patients' faces relative to those of actors/actresses' faces. We agree with Reviewer #1 that there might be individual differences in understanding of the task instructions. This why we further obtained objective estimation of empathy by recording brain activities in response to others' pain as objective estimations of empathy for pain, which provided consistent results across our EEG/fMRI studies. It should be noted that, our EEG/fMRI results replicated previous findings of empathic neural responses in the same neural network and time windows, and our behavioral and EEG/fMRI results together support our conclusion of BOP effects on empathy.

In addition, we conducted VPS analyses to examine specifically how neural activities in the empathy-related regions identified in the previous research (Krishnan et al., 2016, *eLife*) were modulated by beliefs of others’ pain. The results provide further evidence for our hypothesis. We also reported new results of RSA analyses that activities in the brain regions supporting affective sharing (e.g., insula), sensorimotor resonance (e.g., post-central gyrus), and emotion regulation (e.g., lateral frontal cortex) provide intermediate mechanisms underlying variations of subjective feelings of others' pain intensity due to lack of BOP. These were reported in the revision (page 37-39). We believe that, putting all these results together, our paper provides consistent evidence that empathy and altruistic behavior are modulated by BOP.

Additional remarks:– It is not clear to me why the authors refer to the patient-to-patient condition as an enhancement of BOP. The results do indeed suggest that the rating and donation increases, but the manipulation was not meant to enhance the BOP, but to stay stable (an example of enhanced BOP would be if participants would have been told that in the first run they saw actors, but then discovered they were actually patients). Authors cannot define their manipulation based on the results. In all their experiments, one identity remains the same, and only the other changes, therefore BOP is expected to remain the same in one case and decrease (or increase) in the other condition. I would advise to rename their BOP accordingly, and not based on the results.

Thanks for this helpful comment. Indeed, we did not test how BOP changes in the patient-to-patient condition, although it is likely that repetition of the patient identity of faces in the 2^nd^ round test might enhance BOP compared to the 1^st^ round test. As suggested by Reviewer #1 we renamed the patient-to-patient condition as *patient-identity repetition* In Experiments 1 and 5. We also renamed the patient-to-actor/actress condition as *patient-identity change*. *Patient-identity repetition* and *patient-identity change* describe the two conditions objectively. In addition, we discussed possible reasons for increased empathy and donation in the *patient-identity repetition* condition in the revision (page 41-42).

– The authors focus on the N2 components, which makes the manuscript nicely focused. The rationale of this choice is though not very clear, as the literature suggests other components also to be modulated by task instructions (see for instance the following meta-review https://academic.oup.com/scan/article/13/10/1003/5077584?login=true). Authors should justify the focus or report the results for the other components. As mentioned above, authors should also be careful in equating N2 as pain-specific component.

Thanks for this comment. It should be clarified that our manipulation of beliefs modulates neural activity in the P2 (but not N2) time window in response to others' pain expressions. We believe that Reviewer #1 commented on the P2 results here. We cited Coll's paper of a meta-analysis of ERP investigations of pain empathy in the revision. In particular, we clarified in the revised Introduction (page 8) that the following ERP findings are particularly related to the current work, i.e., pain compared to neutral expressions increased the amplitude of a positive component at 128–188 ms (P2) after face onset over the frontal/central regions, and the P2 component is associated with the ACC activity in response to others' pain expressions (Sheng and Han, 2012; Sheng et al., 2013; 2016; Han et al., 2016; Li and Han, 2019). Moreover, the P2 amplitudes in response to others' pain expressions positively predicted subjective feelings of own unpleasantness induced by others' pain and self-report of one's own empathy traits (e.g., Sheng and Han, 2012). These brain imaging findings suggest the P2 component as a good candidate for objective measures of empathy when viewing others' pain expressions.

– The ANOVA on N2 shows a nice Identity x Expression interaction, which the authors interpret as a reduction of empathic neural responses, as the difference between pain and neutral is bigger while watching patient's facial expression. Beside the already mentioned problem of taking N2 as a marker of pain empathy, by looking at the graph I seem to notice that the change in the difference, could be more driven by a change in the N2 amplitude in the response to neutral stimuli (increase of N2 amplitude while watching the actors), rather than a reduction of N2 to the painful stimuli, which seems to stay stable across conditions. If this were indeed the case, it would again suggest that what might have been modulated is not per-se the specific response to pain, but to something like saliency (someone not saying the truth becomes salient) or trust.

Thanks for this comment. Again, we believe Reviewer #1 mentioned P2 (but not N2) component here. We'd like to clarify the results by giving the exact numbers of the P2 amplitudes in different conditions. In the Supplementary tables we reported the mean P2 amplitudes in different conditions in Exp. 3 (Supplementary Table 3_2) and Exp. 5 (Supplementary Table 5_1). In Exp.3 the P2 amplitudes to pain expressions of patient vs. actor faces were 3.7 vs. 3.1 µV, and to neutral expressions of patient vs. actor faces were 2.7 vs. 2.9 µV. In Exp.5 the P2 amplitudes to pain expressions of patient vs. actor faces were 3.9 vs. 3.3 µV, and to neutral expressions of patient vs. actor faces were 3.0 vs. 3.2 µV. These results indicate that modulation of the differential P2 amplitudes to pain (vs. neutral) expressions by BOP was actually more driven by the change of the P2 amplitude in the response to painful rather than neutral stimuli. Besides, in the revised Introduction (page 7), we further clarified how brain imaging studies identified empathic neural responses and that brain responses to perceived non-painful stimuli applied to others or neutral expressions were necessarily collected in previous neuroimaging studies to control empathy-unrelated perceptual or motor processes.

– The manipulation of the team-identity in Exp4 is weak. Of course, it is good that it does not change the believe of how much the different team-mate should feel pain (which is what needed indeed for such a control experiment), but unfortunately it also doesn't create a systematic bias of any kind. As it stands, it boils down to a lable permutation of the identities, which one would not expect to play any modulatory role on any beliefs. If the goal was to test the specificity for pain of the BOP, the authors should have chosen a believe manipulation that was not affecting the pain domain, but another domain, for instance choosing a team the participants was a fan of, and a team that participants do not care about or does not like. By choosing two irrelevant teams, the observed lack of an interaction is trivial, and the authors cannot conclude that BOP is necessary for the modulation of empathic activity.

Thanks for this comment. We're sorry for not making clear the goal of Exp. 4 in the original submission. The goal of Exp. 4 was not to test the specificity for pain of the BOP (we included Supplementary Experiment 1 to address this issue in the revision). In Exp. 3 we showed evidence that BOP modulated the P2 amplitude to pain (vs. neutral) expressions. However, the procedure to manipulate BOP by assigning patient or actor/actress identities to faces in Exp. 3 consisted of multiple processes, including learning, memory and recognition of face identities, assignment to different social groups (e.g., patient or actor groups), etc. Therefore, in Exp. 4, we examined whether these processes including learning, memory, identity recognition, etc. even without BOP differences produced through these processes, would be able to interpret the results in Exp. 3. To this end, we employed the learning procedure used in Exp. 3 to generate two categories of faces (from Tiger or Lion teams). It should be noted that the learning procedure was the same in Exp. 3 and Exp. 4. The only difference between the two experiments was that patient/actor/actress identifies activated different beliefs regarding authenticity of painful emotional states of faces with pain expression, whereas Tiger or Lion team identifies did not. The results of Exp. 3 and Exp. 4 together indicate that the processes involved in the learning procedure were not sufficient to affect the P2 amplitudes in response to pain (vs. neutral) expressions. However, the difference in BOP activated by patient/actor/actress identifies were necessary for the P2 modulations because the same learning procedure that did not change BOP was unable to modulate the P2 amplitudes in response to pain (vs. neutral) expressions. We clarified these in the revised Results section (page 27-28).

– In Exp6 it is not clear why the authors do not present the results of the interaction, which is a central contrast throughout the manuscript. Again, the authors lack evidence for a modulation of pain-empathy. What the authors could do to tap into the questions of whether their BOP modulation affects empathy for pain, is to use some form of multivariate pattern that has higher specificity for pain processing. One way the author could try to use to be able to say something about the involvement of empathy for pain is by using the vicarious pain signature maps (Somatic and vicarious pain are represented by dissociable multivariate brain patterns eLife (elifesciences.org)). Author could multiply the VPS maps with their β weights from the SPM model for the four conditions: patient-actor pain, patient-actor neutral, patient-patient pain, patient-patient neutral, and run an ANOVA on the resulting values. In this way they could basically test the interaction within a network that has been reliably associated with pain-empathy, and the validity of which has been tested across experiments.

We thank Reviewer #1 for the suggestion of using the VPS maps to explore the BOP effects on brain responses to others' pain. This is an excellent idea to test how empathic neural responses are modulated by BOP. Accordingly, we calculated the VPS pattern responses to pain expressions of faces with patient or actor/actress identities for each single-participant using the VPS map in response to perceived noxious stimulation of body limbs (Krishnan et al., 2016) or in response to both perceived noxious stimulation of body limbs and painful facial expressions (Zhou et al., 2020). The results of both analyses showed decreased activities in the VPS pattern in response to pain expressions of faces with actor/actress than patient identities. These results were reported in the revision (page 40). We appreciate very much for Reviewer #1's insightful suggestion.

We'd like to clarify that we did conducted a whole-brain interaction (Identity (patient vs. actor) x Expression (pain vs. neutral)) analysis but did not find significant interaction, and we reported this in the revised Results section (page 38). Since this univariate analysis failed to get reliable results, we further conducted a multivariate representational similarity analysis (RSA) of BOLD signals to assess neural correlates of BOP effects on subjective feeling of others' pain. This analysis revealed significant activations in the left anterior insula and inferior parietal cortex, the right anterior insula/frontal cortex, superior temporal gyrus, inferior post-central gyrus, and superior parietal cortex. We reported the RSA results in the revision (page 38-39) which revealed brain regions in which patterns of neural responses predicted patterns of self-reports of subjective feeling of patients' and actors' pain. The univariate whole-brain interaction (Identity (patient vs. actor) x Expression (pain vs. neutral)) analysis only considered stimulus attributes (e.g., pain or non-pain, patient vs. actor). RSA, however, considered both stimulus attributes and participants' feelings of others' pain in different conditions. This is possibly why RSA works better in uncovering the neural activities that correspond to distinct subjective feelings of patients' and actors/actresses’ pain. Taken together, our fMRI results supplement our EEG results by revealing the key nodes of the empathy network in which activities were activated in correspondence with distinct patterns of subjective feelings of patients' and actors/actresses’ pain. These provide additional evidence for BOP effects on empathic brain responses.

Here below a list of some of the statements unsupported by the data:– "Manipulated BOP changes empathy and altruistic behavior". As I tried to explain, there is no clear measure of empathy, due to the interpretability of the tasks from the participants' point of view. The statement is correctly supported in the case of donation. The concern is only about empathy.

Thanks for this comment. As we clarified in the revised Introduction (page 6-7), previous studies of empathy employed both subjective and objective estimation of empathy for others' pain. Subjective estimation of empathy for pain depends on collection of self report of the degree of others' pain and ones' own unpleasant feelings when observing others' suffering. Objective estimation of empathy for pain relies on collection of brain activities (using fMRI, EEG, MEG, etc.) that differentially respond to painful or non-painful stimuli applied to others. fMRI studies revealed greater activations in the ACC, AI, and sensorimotor cortices in response to painful vs. non-painful stimuli applied to others. EEG studies showed that event-related potentials (ERPs) in response to perceived painful stimulations applied to others' body parts elicited neural responses that differentiated between painful and neutral stimuli over the frontal region as early as 140 ms after stimulus onset (Fan and Han, 2008; see Coll, 2018 for review). Moreover, the mean ERP amplitudes at 140–180 ms predicted subjective reports of others' pain and ones' own unpleasantness. Particularly related to the current study, previous research showed that pain compared to neutral expressions increased the amplitude of the frontal P2 component at 128–188 ms after face (Sheng and Han, 2012; Sheng et al., 2013; 2016; Han et al., 2016; Li and Han, 2019) and the P2 amplitudes in response to others' pain expressions positively predicted subjective feelings of own unpleasantness induced by others' pain and self-report of one's own empathy traits (e.g., Sheng and Han, 2012). These brain imaging findings indicate that brain responses to others' pain can (1) differentiate others' painful or non-painful emotional states to support *understanding* of others' pain and (2) predict subjective feelings of others' pain and one's own unpleasantness induced by others' pain to support *sharing* of others' painful feelings. These findings provide effective subjective and objective measures of empathy that allow the current study to investigate neural mechanisms underlying modulations of empathy and altruism by beliefs of others’ pain. Similarly, our current study employed both subjective and objective measures of empathy and showed consistent evidence for modulation of empathy by BOP. We also discussed these in the revised Discussion (page 41-44).

– "Intrinsic BOP predicts empathy and …". Again it is not clear what participants are reporting and interpreting after they know that the treatment did not have an effect (pity, despair, helplessness).

Thanks for this comment. We'd like to clarify that, in all the experiments of the current study, we gave clear task instructions to participants for estimation of their subjective feelings of others' pain and own unpleasantness. After the presentation of each photo the participants were asked to evaluate intensity of each patient’s pain by rating on a Likert-type scale ("How painful do you think this person is feeling?", 0 = not painful at all; 10 = extremely painful). The participants were also asked to report how unpleasant they were feeling when they viewed the photo (i.e., own unpleasantness) by rating on a Likert-type scale ("How unpleasant do you feel when viewing this person?" 0 = not unpleasant at all, 10 = extremely unpleasant). These rating tasks were adopted from previous research (Bieri et al., 1990; Jackson et al., 2005; Lamm et al., 2007; Fan and Han, 2008; Sheng and Han, 2012) as subjective measures of understanding and sharing of others’ pain feeling — the two key components of empathy. Because the same task instructions were given in the 1^st^ and 2^nd^ round tests, the participants interpreted the task instructions in the same way in both the 1^st^ and 2^nd^ round tests. Moreover, our study conducted both subjective and objective (i.e., neuroimaging) measures of empathy and the results of the two types of measures complemented each other and were consistent regarding BOP effects on empathy. Thus self-reports of rating scores and brain responses to pain (vs. neutral) expressions indeed provide two types of measures of empathy.

To describe the results of Exp. 2 more precisely, we changed the title of Exp. 2 to "Intrinsic BOP predicts subjective estimation of empathy and altruistic behavior". We appreciate this comment from Reviewer #1 because we agree that other kinds of emotions such as pity, despair, helplessness may influence the self-report results. However, these effects may help to explain why the participants reported greater pain intensity and unpleasantness for the 0% effective target faces but not for the 100% effective target faces. For example, 0% effectiveness of treatment might increase pity for targets whereas 100% effectiveness of treatment would not. Most importantly, empathy changes for 100% effective targets are the key results for testing our hypothesis. We thank again for Reviewer #1 for this comment and included additional discussion in the revision regarding possible changes of other emotions related to 0% effective target faces (page 41-42). Specifically, we discussed alternative emotional explanations of the increased donations to targets without effective treatment in addition to the empathy account.

– "Manipulated BOP changes empathic brain activity". The presence of a N2 is not sufficient to claim that there is empathic brain activity, as N2 can be elicited by salient stimuli as well (even the fact that there is an N2 in response to the neutral stimuli, suggests that N2 might represent other activity than empathic).

Thanks for this comment. We believe Reviewer #1 mentioned P2 (but not N2) component here. Indeed, we agree with Reviewer #1 that the P2 component can be elicited by various types of visual stimuli, and this is why we did not simply take the P2 amplitude in response to pain expressions as a neural marker of empathy. Instead, similar to previous ERP research (Sheng and Han, 2012; Sheng et al., 2013; 2016; Han et al., 2016; Li and Han, 2019; Zhou and Han, 2021), we calculate the difference in the P2 amplitudes in response to pain compared to neutral expressions as a neural marker of empathy for pain. As we mentioned in the revised Introduction, the difference in P2 amplitude in response to pain compared to neutral expressions is associated with self-report of others’ pain (Sheng and Han, 2012). Thus using the difference in P2 amplitude in response to pain compared to neutral expressions, rather than simply using the P2 amplitude in response to pain expressions, to assess empathic neural responses rule out empathy-unrelated processes such as facial structure and social attributes. In the revised Introduction, we made clear this logic for examination of brain activities related to empathy in our brain imaging experiments (page 7 and 9).

– "Together, the results in Experiment 3 and 4 suggest a key role of BOP, but not of other cognitive processes involved in the experimental manipulations, in modulations of neuronal responses to other's pain." The control experiment does not contain any other manipulation of cognitive beliefs, and therefore the authors cannot exclude that other beliefs not related to pain could have equally modulated the N2 component.

Thanks for this comment. We appreciate Reviewer #1's clarification that the control experiment (Exp. 4) did not contain any other manipulation of cognitive beliefs. The key point here is to compare the results in Exp. 4 with those in Exp. 3 rather than examining the results of Exp. 4 independently. Exp. 3 and Exp. 4 engaged similar learning, memory, identity recognition, and other processes but were different in that Exp. 3 manipulated BOP whereas Exp. 4 did not. This difference corresponds to the presence modulations of the P2 amplitudes in response to pain (vs. neutral) expressions in Exp. 3 but not in Exp.4. The results of the two experiments together suggest that the manipulation of BOP in Exp. 3 was necessary for modulations of empathic (P2) responses to pain (vs. neutral) expressions. We are sorry for not making clear the rationale of including Experiment 4. In the revised Results section, we made clear the goal and rationale of Experiment 4 (page 27-28).

– "Our behavioral and neuroimaging results revealed critical functional roles of BOP in modulation of the perception-emotion-behavior reactivity by showing how BOP predicted and affected empathy/empathic brain activity and monetary donation. Our finding provide evidence that BOP constitutes a fundamental cognitive basis for empathy and altruistic behavior." The neuroimaging results revealed that areas involved in cognitive processes are modulated by cognitive manipulation. There is no evidence that empathy-related neuronal activity was modulated by the BOP manipulation. BOP is a successful cognitive manipulation, but what authors cannot conclude that BOP is a fundamental cognitive basis for empathy.

Thanks for this comment. As mentioned above, our results of RSA and VPS analyses revealed additional evidence that empathy-related activity was modulated by BOP. These were reported in the revised Results section (page 37-40). We'd be happy to clarify again that there are both cognitive and affective components of empathy, which are mediated by distinct neural substrates. These have been supported by numerous brain imaging findings. Thus we examined BOP effects on both cognitive and affective processes involved in empathy. Our RSA examined whether activities in the empathic network were associated with the dissimilarity matrix constructed from scores of pain intensity in different conditions. This analysis revealed activation in the insula, suggesting that activity in the affective node of the empathy network is associated with BOP modulations of subjective feelings of others' pain. We agree with Reviewer #1 that "BOP is a fundamental cognitive basis for empathy" is probably a strong conclusion. We thus played down the tone in the revision and made a weaker conclusion by saying that "Our findings suggest that BOP may provide a cognitive basis for empathy and altruistic behavior ".

– "These findings together support that empathy, as either an instant emotional response to other's suffering or a sustained personality trait of understanding and sharing other's pain, is one of the key motivation factors for altruistic behavior." I do not see how the results support such a claim, and there is no measure of sustained personality trait of understanding either.

Thanks for this comment. We feel sorry for not making clear this statement. We clarified in the revision that previous work used questionnaires to estimate empathy as a personality trait and our current study used self-report/neuroimaging to examine instant empathic responses to others' suffering shown in their facial expressions. We modified these sentences in the revised Discussion as " Our results showed that self-report of other’s pain intensity and own unpleasantness elicited by perception of others' pain were able to positively predict altruistic behavior across individuals. Previous research using questionnaire measures of empathy ability found that empathy as a trait is positively correlated with the amount of money shared with others in economic games (Edele et al., 2013; Li et al., 2019). These findings are consistent with the proposition that empathy, as either an instant emotional response to others' suffering (e.g., estimated in our study) or a personality trait (e.g., estimated in Edele et al., (2013) and Li et al., (2019)), plays a key role in driving altruistic behavior (Batson, 1987; Batson et al., 2015; Eisenberg et al., 2010; Hoffman, 2008; Penner et al., 2005) " (page 42-43).

Reviewer #2 (Recommendations for the authors):Experiment 1 follow-up notes:– As I noted in my public review, the "enhanced" responses towards the targets appearing as Patients in both Rounds 1 and 2 could simply be a contrast effect due to the Patient-Actor targets. I think there are a number of ways the authors could rule out this possibility. For example, you could show just 8 patients in Round 1, followed by 16 patients in Round 2 (8 new, 8 returning from Round 1). Alternatively, you could show 8 actors/actresses and 8 patients in Round 1, and then tell participants that all these targets are actually patients in Round 2. My prediction in both cases would be that you wouldn't see an increase in empathy/altruism/etc. for the targets appearing as patients in both rounds in either case.

Thanks again for suggestion. We agree with Reviewer #1 that there might be alternative accounts of our results of empathy rating scores and monetary donations observed in the behavioral experiments. Because our ERP results focused on decreased empathic neural responses to actor/actress compared to patient targets and our paper also focused on the issue of decreased empathy and altruism related to BOP, we did not include results of new experiments to explain the increased empathy rating scores and monetary donations observed in the behavioral experiments. However, we appreciate Reviewer #2's suggestions and included additional discussion in the revised Discussion regarding alternative accounts of the behavioral results (page 41-42). In particular, we noted that "Alternatively, repeatedly confirming patient identity or knowing the failure of medical treatment might activate other emotional responses to target faces such as pity or helplessness, which might also influence altruistic decisions. The increased empathy rating scores and monetary donations might reflect a contrast effect due to rating patient and actor targets alternately. These possible accounts can be clarified in future work by asking participants to report their emotions and performing rating tasks on patient and actor/actress targets in separate block of trials." We believe that future work will clarify these.

– Moreover, I referred to some confusion about the charitable donations in this study. In Round 1, the charitable donation is described like this: "The participants were informed that the amount of one of their decisions would be selected randomly and endowed to a charity organization to help those who suffered from the same disease. " In Round 2, it's described like this: "The participants had to…report how much they would like to donate from the extra bonus payment to each model. The participants were told that an amount of money would be finally selected from their 2^nd^ round donation decisions and presented to a charity organization after the study." I wasn't sure if the difference in wording (e.g., "a charity organization to help those who suffered from the same disease" vs. just "a charity organization") was important and intentional – is it the same disease-specific charity in both rounds? (And is it the same charity for both target types in Round 2 – e.g., both patients and actors?) If the answer to either of these questions is no, more detail is necessary, since this could potentially explain differences in giving across condition. (Also, if participants know that these targets are not actually receiving their money, but rather that a trial will be selected at random to determine their donation, it's a little surprising that participants would vary their responses at all – IF the money goes to the same charity regardless of target type.)

Thanks again for this comment. We made clear in the revised Method session that " The participants were then asked to conduct the 2^nd^ round test in which each photo was presented again with a word below to indicate patient or actor/actress identity of the face in the photo. The participants had to report again pain intensity of each face and how much they would like to donate to the person shown in the photo. The participants were informed that an amount of money would be finally selected randomly from their 2^nd^ round decisions and donated to one of the patients through the same charity organization."(page 56).

Experiment 2 typographical notes:– In lines 274-277, the authors report the results for intensity and unpleasantness ratings and donations in a manner that's a little hard to follow ("One-sample t-tests showed that the z values were significantly smaller than zero for decreased-BOP patients (correlations between changes in belief and pain intensity/unpleasantness/monetary donation: mean {plus minus} s.d. = -0.304 {plus minus} 0.370, -0.277 {plus minus} 0.455 and -0.236 {plus minus} 0.410; t(59) = -6.352, -4.706 277 and -4.465; ps < 0.001; Cohen's d = 0.822, 0.609 and 0.576; 95% CI = (-0.400, -0.208), 278 (-0.394, -0.159), and (-0.342, -0.130))") – I'd recommend splitting up these measures, rather than grouping by statistical value (e.g., all three means, all three p-values, etc.).

Thanks for this suggestion. Accordingly, these measures were reported separately in the revision (page 19-20).

Experiment 4 open-ended question:– This study did get me thinking about the considerable literature on intergroup empathy that uses minimal groups or real-world group affiliations (e.g., soccer teams, etc.) to demonstrate that people show an in-group bias in empathy for pain. How should we conceptualize those findings in terms of BOP? (e.g., Are differential expectancies about pain tolerance/sensitivity/etc. embedded in these group memberships?) Or are those group-based effects supported by fundamentally different mechanisms?

Thanks for this comment. This is a great question, and, indeed, a complicated one. Our previous studies suggest that multiple neurocognitive mechanisms are involved in ingroup bias in empathy for pain such as lack of attention (Sheng and Han, 2012) and earlier group-based categorization of outgroup faces (Zhou et al., 2020). However, there has been behavioral evidence that white individuals who more strongly endorsed false beliefs about biological differences between blacks and whites (e.g., “black people’s skin is thicker than white people’s skin”) reported lower pain ratings for a black (vs. white) target and suggested less accurate treatment recommendations (Hoffman et al., 2016). While these behavioral findings suggest other beliefs may also modulate empathy for others’ pain and relevant altruistic behavior, the underlying brain mechanisms remain unknown. The paradigms developed in the current study may be considered in future research to examine neural underpinnings of the effects of false beliefs on empathy for pain. We include these discussions in the revised Discussion section (page 48-49).